# Fast and Guaranteed Tensor Decomposition via Sketching

**Yining Wang, Hsiao-Yu Tung, Alex Smola**
Machine Learning Department
Carnegie Mellon University, Pittsburgh, PA 15213
{yiningwa,htung}@cs.cmu.edu
alex@smola.org

**Anima Anandkumar**
Department of EECS
University of California Irvine
Irvine, CA 92697
a.anandkumar@uci.edu

## Abstract

Tensor CANDECOMP/PARAFAC (CP) decomposition has wide applications in statistical learning of latent variable models and in data mining. In this paper, we propose fast and randomized tensor CP decomposition algorithms based on sketching. We build on the idea of count sketches, but introduce many novel ideas which are unique to tensors. We develop novel methods for randomized computation of tensor contractions via FFTs, without explicitly forming the tensors. Such tensor contractions are encountered in decomposition methods such as tensor power iterations and alternating least squares. We also design novel colliding hashes for symmetric tensors to further save time in computing the sketches. We then combine these sketching ideas with existing whitening and tensor power iterative techniques to obtain the fastest algorithm on both sparse and dense tensors. The quality of approximation under our method does not depend on properties such as sparsity, uniformity of elements, etc. We apply the method for topic modeling and obtain competitive results.

**Keywords**: Tensor CP decomposition, count sketch, randomized methods, spectral methods, topic modeling

## 1 Introduction

In many data-rich domains such as computer vision, neuroscience and social networks consisting of multi-modal and multi-relational data, tensors have emerged as a powerful paradigm for handling the data deluge. An important operation with tensor data is its decomposition, where the input tensor is decomposed into a succinct form. One of the popular decomposition methods is the CANDECOMP/PARAFAC (CP) decomposition, also known as canonical polyadic decomposition [12, 5], where the input tensor is decomposed into a succinct sum of rank-1 components. The CP decomposition has found numerous applications in data mining [4, 18, 20], computational neuroscience [10, 21], and recently, in statistical learning for latent variable models [1, 30, 28, 6]. For latent variable modeling, these methods yield consistent estimates under mild conditions such as non-degeneracy and require only polynomial sample and computational complexity [1, 30, 28, 6].

Given the importance of tensor methods for large-scale machine learning, there has been an increasing interest in scaling up tensor decomposition algorithms to handle gigantic real-world data tensors [27, 24, 8, 16, 14, 2, 29]. However, the previous works fall short in many ways, as described subsequently. In this paper, we design and analyze efficient randomized tensor methods using ideas from sketching [23]. The idea is to maintain a low-dimensional sketch of an input tensor and then perform implicit tensor decomposition using existing methods such as tensor power updates, alternating least squares or online tensor updates. We obtain the fastest decomposition methods for both sparse and dense tensors. Our framework can easily handle modern machine learning applications with billions of training instances, and at the same time, comes with attractive theoretical guarantees.

Our main contributions are as follows:

**Efficient tensor sketch construction:**    We propose efficient construction of tensor sketches when the input tensor is available in factored forms such as in the case of empirical moment tensors, where the factor components correspond to rank-1 tensors over individual data samples. We construct the tensor sketch via efficient FFT operations on the component vectors. Sketching each rank-1 component takes $O(n + b \log b)$ operations where $n$ is the tensor dimension and $b$ is the sketch length. This is much faster than the $O(n^p)$ complexity for brute force computations of a $p$th-order tensor. Since empirical moment tensors are available in the factored form with $N$ components, where $N$ is the number of samples, it takes $O((n + b \log b)N)$ operations to compute the sketch.

**Implicit tensor contraction computations:**    Almost all tensor manipulations can be expressed in terms of tensor *contractions*, which involves multilinear combinations of different tensor *fibres* [19]. For example, tensor decomposition methods such as tensor power iterations, alternating least squares (ALS), whitening and online tensor methods all involve tensor contractions. We propose a highly efficient method to directly compute the tensor contractions without forming the input tensor explicitly. In particular, given the sketch of a tensor, each tensor contraction can be computed in $O(n + b \log b)$ operations, regardless of order of the source and destination tensors. This significantly accelerates the brute-force implementation that requires $O(n^p)$ complexity for $p$th-order tensor contraction. In addition, in many applications, the input tensor is not directly available and needs to be computed from samples, such as the case of empirical moment tensors for spectral learning of latent variable models. In such cases, our method results in huge savings by combining implicit tensor contraction computation with efficient tensor sketch construction.

**Novel colliding hashes for symmetric tensors:**    When the input tensor is symmetric, which is the case for empirical moment tensors that arise in spectral learning applications, we propose a novel colliding hash design by replacing the Boolean ring with the complex ring $\mathbb{C}$ to handle multiplicities. As a result, it makes the sketch building process much faster and avoids repetitive FFT operations. Though the computational complexity remains the same, the proposed colliding hash design results in significant speed-up in practice by reducing the actual number of computations.

**Theoretical and empirical guarantees:**    We show that the quality of the tensor sketch does not depend on sparseness, uniform entry distribution, or any other properties of the input tensor. On the other hand, previous works assume specific settings such as sparse tensors [24, 8, 16], or tensors having entries with similar magnitude [27]. Such assumptions are unrealistic, and in practice, we may have both dense and spiky tensors, for example, unordered word trigrams in natural language processing. We prove that our proposed randomized method for tensor decomposition does not lead to any significant degradation of accuracy.

Experiments on synthetic and real-world datasets show highly competitive results. We demonstrate a 10x to 100x speed-up over exact methods for decomposing dense, high-dimensional tensors. For topic modeling, we show a significant reduction in computational time over existing spectral LDA implementations with small performance loss. In addition, our proposed algorithm outperforms collapsed Gibbs sampling when running time is constrained. We also show that if a Gibbs sampler is initialized with our output topics, it converges within several iterations and outperforms a randomly initialized Gibbs sampler run for much more iterations. Since our proposed method is efficient and avoids local optima, it can be used to accelerate the slow burn-in phase in Gibbs sampling.

**Related Works:**    There have been many works on deploying efficient tensor decomposition methods [27, 24, 8, 16, 14, 2, 29]. Most of these works except [27, 2] implement the alternating least squares (ALS) algorithm [12, 5]. However, this is extremely expensive since the ALS method is run in the input space, which requires $O(n^3)$ operations to execute one least squares step on an $n$-dimensional (dense) tensor. Thus, they are only suited for extremely sparse tensors.

An alternative method is to first reduce the dimension of the input tensor through procedures such as *whitening* to $O(k)$ dimension, where $k$ is the tensor rank, and then carry out ALS in the dimension-reduced space on $k \times k \times k$ tensor [13]. This results in significant reduction of computational complexity when the rank is small ($k \ll n$). Nonetheless, in practice, such complexity is still prohibitively high as $k$ could be several thousands in many settings. To make matters even worse, when the tensor corresponds to empirical moments computed from samples, such as in spectral learning of latent variable models, it is actually much slower to construct the reduced dimension

Table 1: Summary of notations. See also Appendix F.

| Variables | Operator | Meaning | Variables | Operator | Meaning |
|---|---|---|---|---|---|
| $\boldsymbol{a}, \boldsymbol{b} \in \mathbb{C}^n$ | $\boldsymbol{a} \circ \boldsymbol{b} \in \mathbb{C}^n$ | Element-wise product | $\boldsymbol{a} \in \mathbb{C}^n$ | $\boldsymbol{a}^{\otimes 3} \in \mathbb{C}^{n \times n \times n}$ | $\boldsymbol{a} \otimes \boldsymbol{a} \otimes \boldsymbol{a}$ |
| $\boldsymbol{a}, \boldsymbol{b} \in \mathbb{C}^n$ | $\boldsymbol{a} * \boldsymbol{b} \in \mathbb{C}^n$ | Convolution | $\mathbf{A}, \mathbf{B} \in \mathbb{C}^{n \times m}$ | $\mathbf{A} \odot \mathbf{B} \in \mathbb{C}^{n^2 \times m}$ | Khatri-Rao product |
| $\boldsymbol{a}, \boldsymbol{b} \in \mathbb{C}^n$ | $\boldsymbol{a} \otimes \boldsymbol{b} \in \mathbb{C}^{n \times n}$ | Tensor product | $\mathbf{T} \in \mathbb{C}^{n \times n \times n}$ | $\mathbf{T}_{(1)} \in \mathbb{C}^{n \times n^2}$ | Mode expansion |

$k \times k \times k$ tensor from training data than to decompose it, since the number of training samples is typically very large. Another alternative is to carry out online tensor decomposition, as opposed to batch operations in the above works. Such methods are extremely fast [14], but can suffer from high variance. The sketching ideas developed in this paper will improve our ability to handle larger sizes of mini-batches and therefore result in reduced variance in online tensor methods.

Another alternative method is to consider a randomized sampling of the input tensor in each iteration of tensor decomposition [27, 2]. However, such methods can be expensive due to I/O calls and are sensitive to the sampling distribution. In particular, [27] employs uniform sampling, which is incapable of handling tensors with spiky elements. Though non-uniform sampling is adopted in [2], it requires an additional pass over the training data to compute the sampling distribution. In contrast, our sketch based method takes only one pass of the data.

## 2 Preliminaries

**Tensor, tensor product and tensor decomposition** A 3rd order tensor [1] $\mathbf{T}$ of dimension $n$ has $n^3$ entries. Each entry can be represented as $\mathbf{T}_{ijk}$ for $i, j, k \in \{1, \cdots, n\}$. For an $n \times n \times n$ tensor $\mathbf{T}$ and a vector $\boldsymbol{u} \in \mathbb{R}^n$, we define two forms of tensor products (contractions) as follows:

$$\mathbf{T}(\boldsymbol{u}, \boldsymbol{u}, \boldsymbol{u}) = \sum_{i,j,k=1}^{n} \mathbf{T}_{i,j,k} \boldsymbol{u}_i \boldsymbol{u}_j \boldsymbol{u}_k; \quad \mathbf{T}(\mathbf{I}, \boldsymbol{u}, \boldsymbol{u}) = \left[ \sum_{j,k=1}^{n} \mathbf{T}_{1,j,k} \boldsymbol{u}_j \boldsymbol{u}_k, \cdots, \sum_{j,k=1}^{n} \mathbf{T}_{n,j,k} \boldsymbol{u}_j \boldsymbol{u}_k \right].$$

Note that $\mathbf{T}(\boldsymbol{u}, \boldsymbol{u}, \boldsymbol{u}) \in \mathbb{R}$ and $\mathbf{T}(\mathbf{I}, \boldsymbol{u}, \boldsymbol{u}) \in \mathbb{R}^n$. For two complex tensors $\mathbf{A}, \mathbf{B}$ of the same order and dimension, its inner product is defined as $\langle \mathbf{A}, \mathbf{B} \rangle := \sum_l \mathbf{A}_l \overline{\mathbf{B}_l}$, where $l$ ranges over all tuples that index the tensors. The Frobenius norm of a tensor is simply $\|\mathbf{A}\|_F = \sqrt{\langle \mathbf{A}, \mathbf{A} \rangle}$.

The *rank-k CP decomposition* of a 3rd-order $n$-dimensional tensor $\mathbf{T} \in \mathbb{R}^{n \times n \times n}$ involves scalars $\{\lambda_i\}_{i=1}^{k}$ and $n$-dimensional vectors $\{\boldsymbol{a}_i, \boldsymbol{b}_i, \boldsymbol{c}_i\}_{i=1}^{k}$ such that the residual $\|\mathbf{T} - \sum_{i=1}^{k} \lambda_i \boldsymbol{a}_i \otimes \boldsymbol{b}_i \otimes \boldsymbol{c}_i\|_F^2$ is minimized. Here $\mathbf{R} = \boldsymbol{a} \otimes \boldsymbol{b} \otimes \boldsymbol{c}$ is a 3rd order tensor defined as $\mathbf{R}_{ijk} = \boldsymbol{a}_i \boldsymbol{b}_j \boldsymbol{c}_k$. Additional notations are defined in Table 1 and Appendix F.

**Robust tensor power method** The method was proposed in [1] and was shown to provably succeed if the input tensor is a noisy perturbation of the sum of $k$ rank-1 tensors whose base vectors are orthogonal. Fix an input tensor $\mathbf{T} \in \mathbb{R}^{n \times n \times n}$, The basic idea is to randomly generate $L$ initial vectors and perform $T$ power update steps: $\hat{\boldsymbol{u}} = \mathbf{T}(\mathbf{I}, \boldsymbol{u}, \boldsymbol{u}) / \|\mathbf{T}(\mathbf{I}, \boldsymbol{u}, \boldsymbol{u})\|_2$. The vector that results in the largest eigenvalue $\mathbf{T}(\boldsymbol{u}, \boldsymbol{u}, \boldsymbol{u})$ is then kept and subsequent eigenvectors can be obtained via deflation. If implemented naively, the algorithm takes $O(kn^3 LT)$ time to run [2], requiring $O(n^3)$ storage. In addition, in certain cases when a second-order moment matrix is available, the tensor power method can be carried out on a $k \times k \times k$ whitened tensor [1], thus improving the time complexity by avoiding dependence on the ambient dimension $n$. Apart from the tensor power method, other algorithms such as Alternating Least Squares (ALS, [12, 5]) and Stochastic Gradient Descent (SGD, [14]) have also been applied to tensor CP decomposition.

**Tensor sketch** *Tensor sketch* was proposed in [23] as a generalization of count sketch [7]. For a tensor $\mathbf{T}$ of dimension $n_1 \times \cdots \times n_p$, random hash functions $h_1, \cdots, h_p : [n] \rightarrow [b]$ with $\Pr_{h_j}[h_j(i) = t] = 1/b$ for every $i \in [n], j \in [p], t \in [b]$ and binary Rademacher variables $\xi_1, \cdots, \xi_p : [n] \rightarrow \{\pm 1\}$, the sketch $s_{\mathbf{T}} : [b] \rightarrow \mathbb{R}$ of tensor $\mathbf{T}$ is defined as

$$s_{\mathbf{T}}(t) = \sum_{H(i_1, \cdots, i_p) = t} \xi_1(i_1) \cdots \xi_p(i_p) \mathbf{T}_{i_1, \cdots, i_p}, \quad (1)$$

where $H(i_1, \cdots, i_p) = (h_1(i_1) + \cdots + h_p(i_p)) \mod b$. The corresponding recovery rule is $\widehat{\mathbf{T}}_{i_1, \cdots, i_p} = \xi_1(i_1) \cdots \xi_p(i_p) s_{\mathbf{T}}(H(i_1, \cdots, i_p))$. For accurate recovery, $H$ needs to be 2-wise independent, which is achieved by independently selecting $h_1, \cdots, h_p$ from a 2-wise independent hash family [26]. Finally, the estimation can be made more robust by the standard approach of taking $B$ independent sketches of the same tensor and then report the median of the $B$ estimates [7].

# 3   Fast tensor decomposition via sketching

In this section we first introduce an efficient procedure for computing sketches of factored or empirical moment tensors, which appear in a wide variety of applications such as parameter estimation of latent variable models. We then show how to run tensor power method directly on the sketch with reduced computational complexity. In addition, when an input tensor is symmetric (i.e., $\mathbf{T}_{ijk}$ the same for all permutations of $i, j, k$) we propose a novel "colliding hash" design, which speeds up the sketch building process. Due to space limits we only consider the robust tensor power method in the main text. Methods and experiments for sketching based ALS are presented in Appendix C.

To avoid confusions, we emphasize that $n$ is used to denote the dimension of the tensor *to be decomposed*, which is not necessarily the same as the dimension of the original data tensor. Indeed, once whitening is applied $n$ could be as small as the intrinsic dimension $k$ of the original data tensor.

## 3.1   Efficient sketching of empirical moment tensors

Sketching a 3rd-order dense $n$-dimensional tensor via Eq. (1) takes $O(n^3)$ operations, which in general cannot be improved because the input size is $\Omega(n^3)$. However, in practice data tensors are usually structured. One notable example is *empirical moment tensors*, which arises naturally in parameter estimation problems of latent variable models. More specifically, an empirical moment tensor can be expressed as $\mathbf{T} = \hat{\mathbb{E}}[\boldsymbol{x}^{\otimes 3}] = \frac{1}{N} \sum_{i=1}^N \boldsymbol{x}_i^{\otimes 3}$, where $N$ is the total number of training data points and $\boldsymbol{x}_i$ is the $i$th data point. In this section we show that computing sketches of such tensors can be made significantly more efficient than the brute-force implementations via Eq. (1). The main idea is to sketch low-rank components of $\mathbf{T}$ efficiently via FFT, a trick inspired by previous efforts on sketching based matrix multiplication and kernel learning [22, 23].

We consider the more generalized case when an input tensor $\mathbf{T}$ can be written as a weighted sum of known rank-1 components: $\mathbf{T} = \sum_{i=1}^N a_i \boldsymbol{u}_i \otimes \boldsymbol{v}_i \otimes \boldsymbol{w}_i$, where $a_i$ are scalars and $\boldsymbol{u}_i, \boldsymbol{v}_i, \boldsymbol{w}_i$ are known $n$-dimensional vectors. The key observation is that the sketch of each rank-1 component $\mathbf{T}_i = \boldsymbol{u}_i \otimes \boldsymbol{v}_i \otimes \boldsymbol{w}_i$ can be efficiently computed by FFT. In particular, $s_{\mathbf{T}_i}$ can be computed as

$$s_{\mathbf{T}_i} = s_{1, \boldsymbol{u}_i} * s_{2, \boldsymbol{v}_i} * s_{3, \boldsymbol{w}_i} = \mathcal{F}^{-1}(\mathcal{F}(s_{1, \boldsymbol{u}_i}) \circ \mathcal{F}(s_{2, \boldsymbol{v}_i}) \circ \mathcal{F}(s_{3, \boldsymbol{w}_i})), \qquad (2)$$

where $*$ denotes convolution and $\circ$ stands for element-wise vector product. $s_{1, \boldsymbol{u}}(t) = \sum_{h_1(i)=t} \xi_1(i) \boldsymbol{u}_i$ is the count sketch of $\boldsymbol{u}$ and $s_{2, \boldsymbol{v}}, s_{3, \boldsymbol{w}}$ are defined similarly. $\mathcal{F}$ and $\mathcal{F}^{-1}$ denote the Fast Fourier Transform (FFT) and its inverse operator. By applying FFT, we reduce the convolution computation into element-wise product evaluation in the Fourier space. Therefore, $s_{\mathbf{T}}$ can be computed using $O(n + b \log b)$ operations, where the $O(b \log b)$ term arises from FFT evaluations. Finally, because the sketching operator is linear (i.e., $s(\sum_i a_i \mathbf{T}_i) = \sum_i a_i s(\mathbf{T}_i)$), $s_{\mathbf{T}}$ can be computed in $O(N(n + b \log b))$, which is much cheaper than brute-force that takes $O(Nn^3)$ time.

## 3.2   Fast robust tensor power method

We are now ready to present the fast robust tensor power method, the main algorithm of this paper. The computational bottleneck of the original robust tensor power method is the computation of two tensor products: $\mathbf{T}(\mathbf{I}, \boldsymbol{u}, \boldsymbol{u})$ and $\mathbf{T}(\boldsymbol{u}, \boldsymbol{u}, \boldsymbol{u})$. A naive implementation requires $O(n^3)$ operations. In this section, we show how to speed up computation of these products. We show that given the sketch of an input tensor $\mathbf{T}$, one can approximately compute both $\mathbf{T}(\mathbf{I}, \boldsymbol{u}, \boldsymbol{u})$ and $\mathbf{T}(\boldsymbol{u}, \boldsymbol{u}, \boldsymbol{u})$ in $O(b \log b + n)$ steps, where $b$ is the hash length.

Before going into details, we explain the key idea behind our fast tensor product computation. For any two tensors $\mathbf{A}, \mathbf{B}$, its inner product $\langle \mathbf{A}, \mathbf{B} \rangle$ can be approximated by [4]

$$\langle \mathbf{A}, \mathbf{B} \rangle \approx \langle s_{\mathbf{A}}, s_{\mathbf{B}} \rangle. \qquad (3)$$

**Algorithm 1** Fast robust tensor power method

---

1: **Input**: noisy symmetric tensor $\bar{\mathbf{T}} = \mathbf{T} + \mathbf{E} \in \mathbb{R}^{n \times n \times n}$; target rank $k$; number of initializations $L$, number of iterations $T$, hash length $b$, number of independent sketches $B$.
2: **Initialization**: $h_j^{(m)}, \xi_j^{(m)}$ for $j \in \{1, 2, 3\}$ and $m \in [B]$; compute sketches $\boldsymbol{s}_{\bar{\mathbf{T}}}^{(m)} \in \mathbb{C}^b$.
3: **for** $\tau = 1$ to $L$ **do**
4:     Draw $\boldsymbol{u}_0^{(\tau)}$ uniformly at random from unit sphere.
5:     **for** $t = 1$ to $T$ **do**
6:         For each $m \in [B], j \in \{2, 3\}$ compute the sketch of $\boldsymbol{u}_{t-1}^{(\tau)}$ using $h_j^{(m)}, \xi_j^{(m)}$ via Eq. (1).
7:         Compute $\boldsymbol{v}^{(m)} \approx \bar{\mathbf{T}}(\mathbf{I}, \boldsymbol{u}_{t-1}^{(\tau)}, \boldsymbol{u}_{t-1}^{(\tau)})$ as follows: first evaluate $\bar{\boldsymbol{s}}^{(m)} = \mathcal{F}^{-1}(\mathcal{F}(\boldsymbol{s}_{\bar{\mathbf{T}}}^{(m)}) \circ \overline{\mathcal{F}(\boldsymbol{s}_{2,\boldsymbol{u}}^{(m)})} \circ \overline{\mathcal{F}(\boldsymbol{s}_{3,\boldsymbol{u}}^{(m)})})$. Set $[\boldsymbol{v}^{(m)}]_i$ as $[\boldsymbol{v}^{(m)}]_i \leftarrow \xi_1(i)[\bar{\boldsymbol{s}}^{(m)}]_{h_1(i)}$ for every $i \in [n]$.
8:         Set $\bar{\boldsymbol{v}}_i \leftarrow \text{med}(\Re(\boldsymbol{v}_i^{(1)}), \cdots, \Re(\boldsymbol{v}_i^{(B)}))^3$. Update: $\boldsymbol{u}_t^{(\tau)} = \bar{\boldsymbol{v}}/\|\bar{\boldsymbol{v}}\|$.
9: **Selection** Compute $\lambda_\tau^{(m)} \approx \bar{\mathbf{T}}(\boldsymbol{u}_T^{(\tau)}, \boldsymbol{u}_T^{(\tau)}, \boldsymbol{u}_T^{(\tau)})$ using $\boldsymbol{s}_{\bar{\mathbf{T}}}^{(m)}$ for $\tau \in [L]$ and $m \in [B]$. Evaluate $\lambda_\tau = \text{med}(\lambda_\tau^{(1)}, \cdots, \lambda_\tau^{(B)})$ and $\tau^* = \text{argmax}_\tau \lambda_\tau$. Set $\hat{\lambda} = \lambda_{\tau^*}$ and $\hat{\boldsymbol{u}} = \boldsymbol{u}_T^{(\tau^*)}$.
10: **Deflation** For each $m \in [B]$ compute sketch $\tilde{\boldsymbol{s}}_{\Delta\mathbf{T}}^{(m)}$ for the rank-1 tensor $\Delta\mathbf{T} = \hat{\lambda}\hat{\boldsymbol{u}}^{\otimes 3}$.
11: **Output**: the eigenvalue/eigenvector pair $(\hat{\lambda}, \hat{\boldsymbol{u}})$ and sketches of the deflated tensor $\bar{\mathbf{T}} - \Delta\mathbf{T}$.

---

Table 2: Computational complexity of sketched and plain tensor power method. $n$ is the tensor dimension; $k$ is the intrinsic tensor rank; $b$ is the sketch length. Per-sketch time complexity is shown.

|  | PLAIN | SKETCH | PLAIN+WHITENING | SKETCH+WHITENING |
|---|---|---|---|---|
| preprocessing: general tensors | - | $O(n^3)$ | $O(kn^3)$ | $O(n^3)$ |
| preprocessing: factored tensors with $N$ components | $O(Nn^3)$ | $O(N(n + b\log b))$ | $O(N(nk + k^3))$ | $O(N(nk + b\log b))$ |
| per tensor contraction time | $O(n^3)$ | $O(n + b\log b)$ | $O(k^3)$ | $O(k + b\log b)$ |

Eq. (3) immediately results in a fast approximation procedure of $\mathbf{T}(\boldsymbol{u}, \boldsymbol{u}, \boldsymbol{u})$ because $\mathbf{T}(\boldsymbol{u}, \boldsymbol{u}, \boldsymbol{u}) = \langle \mathbf{T}, \mathbf{X} \rangle$ where $\mathbf{X} = \boldsymbol{u} \otimes \boldsymbol{u} \otimes \boldsymbol{u}$ is a rank one tensor, whose sketch can be built in $O(n + b\log b)$ time by Eq. (2). Consequently, the product can be approximately computed using $O(n + b\log b)$ operations if the tensor sketch of $\mathbf{T}$ is available. For tensor product of the form $\mathbf{T}(\mathbf{I}, \boldsymbol{u}, \boldsymbol{u})$. The $i$th coordinate in the result can be expressed as $\langle \mathbf{T}, \mathbf{Y}_i \rangle$ where $\mathbf{Y}_i = \boldsymbol{e}_i \otimes \boldsymbol{u} \otimes \boldsymbol{u}$; $\boldsymbol{e}_i = (0, \cdots, 0, 1, 0, \cdots, 0)$ is the $i$th indicator vector. We can then apply Eq. (3) to approximately compute $\langle \mathbf{T}, \mathbf{Y}_i \rangle$ efficiently. However, this method is not completely satisfactory because it requires sketching $n$ rank-1 tensors ($\mathbf{Y}_1$ through $\mathbf{Y}_n$), which results in $O(n)$ FFT evaluations by Eq. (2). Below we present a proposition that allows us to use only $O(1)$ FFTs to approximate $\mathbf{T}(\mathbf{I}, \boldsymbol{u}, \boldsymbol{u})$.

**Proposition 1.** $\langle \boldsymbol{s}_{\mathbf{T}}, \boldsymbol{s}_{1,\boldsymbol{e}_i} * \boldsymbol{s}_{2,\boldsymbol{u}} * \boldsymbol{s}_{3,\boldsymbol{u}} \rangle = \langle \mathcal{F}^{-1}(\mathcal{F}(\boldsymbol{s}_{\mathbf{T}}) \circ \overline{\mathcal{F}(\boldsymbol{s}_{2,\boldsymbol{u}})} \circ \overline{\mathcal{F}(\boldsymbol{s}_{3,\boldsymbol{u}})}), \boldsymbol{s}_{1,\boldsymbol{e}_i} \rangle$.

Proposition 1 is proved in Appendix E.1. The main idea is to "shift" all terms not depending on $i$ to the left side of the inner product and eliminate the inverse FFT operation on the right side so that $\boldsymbol{s}_{\boldsymbol{e}_i}$ contains only one nonzero entry. As a result, we can compute $\mathcal{F}^{-1}(\mathcal{F}(\boldsymbol{s}_{\mathbf{T}}) \circ \overline{\mathcal{F}(\boldsymbol{s}_{2,\boldsymbol{u}})} \circ \overline{\mathcal{F}(\boldsymbol{s}_{3,\boldsymbol{u}})})$ once and read off each entry of $\mathbf{T}(\mathbf{I}, \boldsymbol{u}, \boldsymbol{u})$ in constant time. In addition, the technique can be further extended to symmetric tensor sketches, with details deferred to Appendix B due to space limits. When operating on an $n$-dimensional tensor, The algorithm requires $O(kLT(n + Bb\log b))$ running time (excluding the time for building $\tilde{\boldsymbol{s}}_{\bar{\mathbf{T}}}$) and $O(Bb)$ memory, which significantly improves the $O(kn^3LT)$ time and $O(n^3)$ space complexity over the brute force tensor power method. Here $L, T$ are algorithm parameters for robust tensor power method. Previous analysis shows that $T = O(\log k)$ and $L = \text{poly}(k)$, where $\text{poly}(\cdot)$ is some low order polynomial function. [1]

Finally, Table 2 summarizes computational complexity of sketched and plain tensor power method.

### 3.3 Colliding hash and symmetric tensor sketch

For symmetric input tensors, it is possible to design a new style of tensor sketch that can be built more efficiently. The idea is to design hash functions that deliberately collide symmetric entries, i.e., $(i, j, k)$, $(j, i, k)$, etc. Consequently, we only need to consider entries $\mathbf{T}_{ijk}$ with $i \leq j \leq k$ when building tensor sketches. An intuitive idea is to use the same hash function and Rademacher random variable for each order, that is, $h_1(i) = h_2(i) = h_3(i) =: h(i)$ and $\xi_1(i) = \xi_2(i) = \xi_3(i) =: \xi(i)$.

In this way, all permutations of $(i, j, k)$ will collide with each other. However, such a design has an issue with repeated entries because $\xi(i)$ can only take $\pm 1$ values. Consider $(i, i, k)$ and $(j, j, k)$ as an example: $\xi(i)^2\xi(k) = \xi(j)^2\xi(k)$ with probability 1 even if $i \neq j$. On the other hand, we need $\mathbb{E}[\xi(a)\xi(b)] = 0$ for any pair of distinct 3-tuples $a$ and $b$.

To address the above-mentioned issue, we extend the Rademacher random variables to the complex domain and consider all roots of $z^m = 1$, that is, $\Omega = \{\omega_j\}_{j=0}^{m-1}$ where $\omega_j = e^{i\frac{2\pi j}{m}}$. Suppose $\sigma(i)$ is a Rademacher random variable with $\Pr[\sigma(i) = \omega_i] = 1/m$. By elementary algebra, $\mathbb{E}[\sigma(i)^p] = 0$ whenever $m$ is relative prime to $p$ or $m$ can be divided by $p$. Therefore, by setting $m = 4$ we avoid collisions of repeated entries in a 3rd order tensor. More specifically, The symmetric tensor sketch of a symmetric tensor $\mathbf{T} \in \mathbb{R}^{n \times n \times n}$ can be defined as

$$\tilde{s}_{\mathbf{T}}(t) := \sum_{\tilde{H}(i,j,k)=t} \mathbf{T}_{i,j,k}\sigma(i)\sigma(j)\sigma(k), \tag{4}$$

where $\tilde{H}(i, j, k) = (h(i) + h(j) + h(k)) \mod b$. To recover an entry, we use

$$\widehat{\mathbf{T}}_{i,j,k} = 1/\kappa \cdot \overline{\sigma(i)} \cdot \overline{\sigma(j)} \cdot \overline{\sigma(k)} \cdot \tilde{s}_{\mathbf{T}}(H(i,j,k)), \tag{5}$$

where $\kappa = 1$ if $i = j = k$; $\kappa = 3$ if $i = j$ or $j = k$ or $i = k$; $\kappa = 6$ otherwise. For higher order tensors, the coefficients can be computed via the Young tableaux which characterizes symmetries under the permutation group. Compared to asymmetric tensor sketches, the hash function $h$ needs to satisfy stronger independence conditions because we are using the same hash function for each order. In our case, $h$ needs to be 6-wise independent to make $\tilde{H}$ 2-wise independent. The fact is due to the following proposition, which is proved in Appendix E.1.

**Proposition 2.** *Fix $p$ and $q$. For $h : [n] \to [b]$ define symmetric mapping $\tilde{H} : [n]^p \to [b]$ as $\tilde{H}(i_1, \cdots, i_p) = h(i_1) + \cdots + h(i_p)$. If $h$ is $(pq)$-wise independent then $H$ is $q$-wise independent.*

The symmetric tensor sketch described above can significantly speed up sketch building processes. For a general tensor with $M$ nonzero entries, to build $\tilde{s}_{\mathbf{T}}$ one only needs to consider roughly $M/6$ entries (those $\mathbf{T}_{ijk} \neq 0$ with $i \leq j \leq k$). For a rank-1 tensor $\boldsymbol{u}^{\otimes 3}$, only one FFT is needed to build $\mathcal{F}(\tilde{s})$; in contrast, to compute Eq. (2) one needs at least 3 FFT evaluations.

Finally, in Appendix B we give details on how to seamlessly combine symmetric hashing and techniques in previous sections to efficiently construct and decompose a tensor.

## 4 Error analysis

In this section we provide theoretical analysis on approximation error of both tensor sketch and the fast sketched robust tensor power method. We mainly focus on symmetric tensor sketches, while extension to asymmetric settings is trivial. Due to space limits, all proofs are placed in the appendix.

### 4.1 Tensor sketch concentration bounds

Theorem 1 bounds the approximation error of symmetric tensor sketches when computing $\mathbf{T}(\boldsymbol{u}, \boldsymbol{u}, \boldsymbol{u})$ and $\mathbf{T}(\mathbf{I}, \boldsymbol{u}, \boldsymbol{u})$. Its proof is deferred to Appendix E.2.

**Theorem 1.** *Fix a symmetric real tensor $\mathbf{T} \in \mathbb{R}^{n \times n \times n}$ and a real vector $\boldsymbol{u} \in \mathbb{R}^n$ with $\|\boldsymbol{u}\|_2 = 1$. Suppose $\varepsilon_{1,T}(\boldsymbol{u}) \in \mathbb{R}$ and $\boldsymbol{\varepsilon}_{2,T}(\boldsymbol{u}) \in \mathbb{R}^n$ are estimation errors of $\mathbf{T}(\boldsymbol{u}, \boldsymbol{u}, \boldsymbol{u})$ and $\mathbf{T}(\mathbf{I}, \boldsymbol{u}, \boldsymbol{u})$ using $B$ independent symmetric tensor sketches; that is, $\varepsilon_{1,T}(\boldsymbol{u}) = \widehat{\mathbf{T}}(\boldsymbol{u}, \boldsymbol{u}, \boldsymbol{u}) - \mathbf{T}(\boldsymbol{u}, \boldsymbol{u}, \boldsymbol{u})$ and $\boldsymbol{\varepsilon}_{2,T}(\boldsymbol{u}) = \widehat{\mathbf{T}}(\mathbf{I}, \boldsymbol{u}, \boldsymbol{u}) - \mathbf{T}(\mathbf{I}, \boldsymbol{u}, \boldsymbol{u})$. If $B = \Omega(\log(1/\delta))$ then with probability $\geq 1 - \delta$ the following error bounds hold:*

$$\left|\varepsilon_{1,T}(\boldsymbol{u})\right| = O(\|\mathbf{T}\|_F/\sqrt{b}); \quad \left|[\boldsymbol{\varepsilon}_{2,T}(\boldsymbol{u})]_i\right| = O(\|\mathbf{T}\|_F/\sqrt{b}), \quad \forall i \in \{1, \cdots, n\}. \tag{6}$$

*In addition, for any fixed $\boldsymbol{w} \in \mathbb{R}^n$, $\|\boldsymbol{w}\|_2 = 1$ with probability $\geq 1 - \delta$ we have*

$$\langle \boldsymbol{w}, \boldsymbol{\varepsilon}_{2,T}(\boldsymbol{u}) \rangle^2 = O(\|\mathbf{T}\|_F^2/b). \tag{7}$$

### 4.2 Analysis of the fast tensor power method

We present a theorem analyzing robust tensor power method with tensor sketch approximations. A more detailed theorem statement along with its proof can be found in Appendix E.3.

**Theorem 2.** *Suppose $\bar{\mathbf{T}} = \mathbf{T} + \mathbf{E} \in \mathbb{R}^{n \times n \times n}$ where $\mathbf{T} = \sum_{i=1}^k \lambda_i \boldsymbol{v}_i^{\otimes 3}$ with an orthonormal basis $\{\boldsymbol{v}_i\}_{i=1}^k$, $\lambda_1 > \cdots > \lambda_k > 0$ and $\|\mathbf{E}\| = \epsilon$. Let $\{(\hat{\lambda}_i, \hat{\boldsymbol{v}}_i)\}_{i=1}^k$ be the eigen-*

Table 3: Squared residual norm on top 10 recovered eigenvectors of 1000d tensors and running time (excluding I/O and sketch building time) for plain (exact) and sketched robust tensor power methods. Two vectors are considered mismatch (wrong) if $\|\boldsymbol{v} - \hat{\boldsymbol{v}}\|_2^2 > 0.1$. A extended version is shown as Table 5 in Appendix A.

| | | Residual norm | | | | | No. of wrong vectors | | | | | Running time (min.) | | | | |
|---|---|---|---|---|---|---|---|---|---|---|---|---|---|---|---|---|
| | $\log_2(b)$: | 12 | 13 | 14 | 15 | 16 | 12 | 13 | 14 | 15 | 16 | 12 | 13 | 14 | 15 | 16 |
| $\sigma = .01$ | $B = 20$ | .40 | .19 | .10 | **.09** | .08 | 8 | 6 | 3 | **0** | 0 | .85 | 1.6 | 3.5 | **7.4** | 16.6 |
| | $B = 30$ | .26 | .10 | .09 | .08 | .07 | 7 | 5 | 2 | 0 | 0 | 1.3 | 2.4 | 5.3 | 11.3 | 24.6 |
| | $B = 40$ | .17 | .10 | **.08** | .08 | .07 | 7 | 4 | **0** | 0 | 0 | 1.8 | 3.3 | **7.3** | 15.2 | 33.0 |
| | Exact | .07 | | | | | 0 | | | | | 293.5 | | | | |

Table 4: Negative log-likelihood and running time (min) on the *large* Wikipedia dataset for 200 and 300 topics.

| $k$ | | like. | time | $\log_2 b$ | iters | $k$ | like. | time | $\log_2 b$ | iters |
|---|---|---|---|---|---|---|---|---|---|---|
| 200 | Spectral | 7.49 | **34** | 12 | - | 300 | 7.39 | **56** | 13 | - |
| | Gibbs | 6.85 | 561 | - | 30 | | 6.38 | 818 | - | 30 |
| | Hybrid | **6.77** | 144 | 12 | 5 | | **6.31** | 352 | 13 | 10 |

*value/eigenvector pairs obtained by Algorithm 1. Suppose $\epsilon = O(1/(\lambda_1 n))$, $T = \Omega(\log(n/\delta) + \log(1/\epsilon)\max_i \lambda_i/(\lambda_i - \lambda_{i-1}))$ and $L$ grows linearly with $k$. Assume the randomness of the tensor sketch is independent among tensor product evaluations. If $B = \Omega(\log(n/\delta))$ and $b$ satisfies*

$$b = \Omega\left(\max\left\{\frac{\epsilon^{-2}\|\mathbf{T}\|_F^2}{\Delta(\boldsymbol{\lambda})^2}, \frac{\delta^{-4}n^2\|\mathbf{T}\|_F^2}{r(\boldsymbol{\lambda})^2\lambda_1^2}\right\}\right) \tag{8}$$

*where $\Delta(\boldsymbol{\lambda}) = \min_i(\lambda_i - \lambda_{i-1})$ and $r(\boldsymbol{\lambda}) = \max_{i,j>i}(\lambda_i/\lambda_j)$, then with probability $\geq 1 - \delta$ there exists a permutation $\pi$ over $[k]$ such that*

$$\|\boldsymbol{v}_{\pi(i)} - \hat{\boldsymbol{v}}_i\|_2 \leq \epsilon, \quad |\lambda_{\pi(i)} - \hat{\lambda}_i| \leq \lambda_i\epsilon/2, \quad \forall i \in \{1, \cdots, k\} \tag{9}$$

*and $\|\mathbf{T} - \sum_{i=1}^k \hat{\lambda}_i\hat{\boldsymbol{v}}_i^{\otimes 3}\| \leq c\epsilon$ for some constant c.*

Theorem 1 shows that the sketch length $b$ can be set as $o(n^3)$ to provably approximately decompose a 3rd-order tensor with dimension $n$. Theorem 1 together with time complexity comparison in Table 2 shows that the sketching based fast tensor decomposition algorithm has better computational complexity over brute-force implementation. One potential drawback of our analysis is the assumption that sketches are independently built for each tensor product (contraction) evaluation. This is an artifact of our analysis and we conjecture that it can be removed by incorporating recent development of differentially private adaptive query framework [9].

## 5 Experiments

We demonstrate the effectiveness and efficiency of our proposed sketch based tensor power method on both synthetic tensors and real-world topic modeling problems. Experimental results involving the fast ALS method are presented in Appendix C.3. All methods are implemented in C++ and tested on a single machine with 8 Intel X5550@2.67Ghz CPUs and 32GB memory. For synthetic tensor decomposition we use only a single thread; for fast spectral LDA 8 to 16 threads are used.

### 5.1 Synthetic tensors

In Table 5 we compare our proposed algorithms with exact decomposition methods on synthetic tensors. Let $n = 1000$ be the dimension of the input tensor. We first generate a random *orthonormal* basis $\{\boldsymbol{v}_i\}_{i=1}^n$ and then set the input tensor $\mathbf{T}$ as $\mathbf{T} = \text{normalize}(\sum_{i=1}^n \lambda_i\boldsymbol{v}_i^{\otimes 3}) + \mathbf{E}$, where the eigenvalues $\lambda_i$ satisfy $\lambda_i = 1/i$. The normalization step makes $\|\mathbf{T}\|_F^2 = 1$ before imposing noise. The Gaussian noise matrix $\mathbf{E}$ is symmetric with $\mathbf{E}_{ijk} \sim \mathcal{N}(0, \sigma/n^{1.5})$ for $i \leq j \leq k$ and noise-to-signal level $\sigma$. Due to time constraints, we only compare the recovery error and running time on the top 10 recovered eigenvectors of the full-rank input tensor $\mathbf{T}$. Both $L$ and $T$ are set to 30. Table 3 shows that our proposed algorithms achieve reasonable approximation error within a few minutes, which is much faster then exact methods. A complete version (Table 5) is deferred to Appendix A.

### 5.2 Topic modeling

We implement a fast spectral inference algorithm for Latent Dirichlet Allocation (LDA [3]) by combining tensor sketching with existing whitening technique for dimensionality reduction. Implemen-

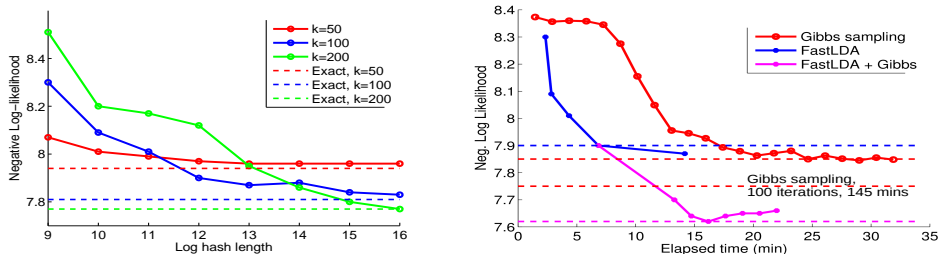

Figure 1: Left: negative log-likelihood for fast and exact tensor power method on Wikipedia dataset. Right: negative log-likelihood for collapsed Gibbs sampling, fast LDA and Gibbs sampling using Fast LDA as initialization.

tation details are provided in Appendix D. We compare our proposed fast spectral LDA algorithm with baseline spectral methods and collapsed Gibbs sampling (using GibbsLDA++ [25] implementation) on two real-world datasets: Wikipedia and Enron. Dataset details are presented in A Only the most frequent $V$ words are kept and the vocabulary size $V$ is set to 10000. For the robust tensor power method the parameters are set to $L = 50$ and $T = 30$. For ALS we iterate until convergence, or a maximum number of 1000 iterations is reached. $\alpha_0$ is set to 1.0 and $B$ is set to 30.

Obtained topic models $\mathbf{\Phi} \in \mathbb{R}^{V \times K}$ are evaluated on a held-out dataset consisting of 1000 documents randomly picked out from training datasets. For each testing document $d$, we fit a topic mixing vector $\hat{\boldsymbol{\pi}}_d \in \mathbb{R}^K$ by solving the following optimization problem: $\hat{\boldsymbol{\pi}}_d = \operatorname{argmin}_{\|\boldsymbol{\pi}\|_1 = 1, \boldsymbol{\pi} \geq \mathbf{0}} \|\boldsymbol{w}_d - \mathbf{\Phi}\boldsymbol{\pi}\|_2$, where $\boldsymbol{w}_d$ is the empirical word distribution of document $d$. The per-document log-likelihood is then defined as $\mathcal{L}_d = \frac{1}{n_d} \sum_{i=1}^{n_d} \ln p(w_{di})$, where $p(w_{di}) = \sum_{k=1}^{K} \hat{\boldsymbol{\pi}}_k \mathbf{\Phi}_{w_{di},k}$. Finally, the average $\mathcal{L}_d$ over all testing documents is reported.

Figure 1 left shows the held-out negative log-likelihood for fast spectral LDA under different hash lengths $b$. We can see that as $b$ increases, the performance approaches the exact tensor power method because sketching approximation becomes more accurate. On the other hand, Table 6 shows that fast spectral LDA runs much faster than exact tensor decomposition methods while achieving comparable performance on both datasets.

Figure 1 right compares the convergence of collapsed Gibbs sampling with different number of iterations and fast spectral LDA with different hash lengths on Wikipedia dataset. For collapsed Gibbs sampling, we set $\alpha = 50/K$ and $\beta = 0.1$ following [11]. As shown in the figure, fast spectral LDA achieves comparable held-out likelihood while running faster than collapsed Gibbs sampling. We further take the dictionary $\mathbf{\Phi}$ output by fast spectral LDA and use it as initializations for collapsed Gibbs sampling (the word topic assignments $\boldsymbol{z}$ are obtained by 5-iteration Gibbs sampling, with the dictionary $\mathbf{\Phi}$ fixed). The resulting Gibbs sampler converges much faster: with only 3 iterations it already performs much better than a randomly initialized Gibbs sampler run for 100 iterations, which takes 10x more running time.

We also report performance of fast spectral LDA and collapsed Gibbs sampling on a larger dataset in Table 4. The dataset was built by crawling 1,085,768 random Wikipedia pages and a held-out evaluation set was built by randomly picking out 1000 documents from the dataset. Number of topics $k$ is set to 200 or 300, and after getting topic dictionary $\mathbf{\Phi}$ from fast spectral LDA we use 2-iteration Gibbs sampling to obtain word topic assignments $\boldsymbol{z}$. Table 4 shows that the hybrid method (i.e., collapsed Gibbs sampling initialized by spectral LDA) achieves the best likelihood performance in a much shorter time, compared to a randomly initialized Gibbs sampler.

## 6  Conclusion

In this work we proposed a sketching based approach to efficiently compute tensor CP decomposition with provable guarantees. We apply our proposed algorithm on learning latent topics of unlabeled document collections and achieve significant speed-up compared to vanilla spectral and collapsed Gibbs sampling methods. Some interesting future directions include further improving the sample complexity analysis and applying the framework to a broader class of graphical models.

**Acknowledgement:**  Anima Anandkumar is supported in part by the Microsoft Faculty Fellowship and the Sloan Foundation. Alex Smola is supported in part by a Google Faculty Research Grant.

## Footnotes

[1] Though we mainly focus on 3rd order tensors in this work, extension to higher order tensors is easy.

[2] $L$ is usually set to be a linear function of $k$ and $T$ is logarithmic in $n$; see Theorem 5.1 in [1].

[3]$\Re(\cdot)$ denotes the real part of a complex number. $\mathrm{med}(\cdot)$ denotes the median.

[4]All approximations will be theoretically justified in Section 4 and Appendix E.2.

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
