[Supplementary Material]

# Appendix A  Supplementary experimental results

The Wikipedia dataset is built by crawling all documents in all subcategories within 3 layers below the *science* category. The Enron dataset is from the Enron email corpus [17]. After usual cleaning steps, the Wikipedia dataset has $114,274$ documents with an average $512$ words per document; the Enron dataset has $186,501$ emails with average $91$ words per email.

Table 5: Squared residual norm on top 10 recovered eigenvectors of 1000d tensors and running time (excluding I/O and sketch building time) for plan (exact) and sketched robust tensor power methods. Two vectors are considered mismatched (wrong) if $\|\boldsymbol{v} - \hat{\boldsymbol{v}}\|_2^2 > 0.1$.

| | | Residual norm | | | | | No. of wrong vectors | | | | | Running time (min.) | | | | |
|---|---|---|---|---|---|---|---|---|---|---|---|---|---|---|---|---|
| | $\log_2(b)$: | 12 | 13 | 14 | 15 | 16 | 12 | 13 | 14 | 15 | 16 | 12 | 13 | 14 | 15 | 16 |
| $\sigma = .01$ | $B = 20$ | .40 | .19 | .10 | **.09** | .08 | 8 | 6 | 3 | **0** | 0 | .85 | 1.6 | 3.5 | **7.4** | 16.6 |
| | $B = 30$ | .26 | .10 | .09 | .08 | .07 | 7 | 5 | 2 | 0 | 0 | 1.3 | 2.4 | 5.3 | 11.3 | 24.6 |
| | $B = 40$ | .17 | .10 | **.08** | .08 | .07 | 7 | 4 | **0** | 0 | 0 | 1.8 | 3.3 | **7.3** | 15.2 | 33.0 |
| | Exact | .07 | | | | | 0 | | | | | 293.5 | | | | |
| $\sigma = .1$ | $B = 20$ | .52 | 3.1 | .21 | **.18** | .17 | 8 | 7 | 4 | **0** | 0 | .84 | 1.6 | 3.5 | **7.5** | 16.8 |
| | $B = 30$ | 4.0 | .24 | .19 | .17 | .16 | 7 | 5 | 3 | 0 | 0 | 1.3 | 2.5 | 5.4 | 11.6 | 26.2 |
| | $B = 40$ | .30 | .22 | **.18** | .17 | .16 | 7 | 4 | **0** | 0 | 0 | 1.8 | 3.3 | **7.3** | 15.5 | 33.5 |
| | Exact | .16 | | | | | 0 | | | | | 271.8 | | | | |

Table 6: Selected negative log-likelihood and running time (min) for fast and exact spectral methods on Wikipedia (top) and Enron (bottom) datasets.

| | | $k = 50$ | | | $k = 100$ | | | $k = 200$ | | |
|---|---|---|---|---|---|---|---|---|---|---|
| | | Fast RB | RB | ALS | Fast RB | RB | ALS | Fast RB | RB | ALS |
| Wiki. | like. | 8.01 | **7.94** | 8.16 | 7.90 | **7.81** | 7.93 | 7.86 | **7.77** | 7.89 |
| | time | **2.2** | 97.7 | 2.4 | **6.8** | 135 | 29.3 | **57.3** | 423 | 677 |
| | $\log_2 b$ | 10 | - | - | 12 | - | - | 14 | - | - |
| Enron | like. | 8.31 | **8.28** | 8.22 | 8.18 | **8.09** | 8.30 | 8.26 | **8.18** | 8.27 |
| | time | **2.4** | 45.8 | 5.2 | **3.7** | 93.9 | 40.6 | **6.4** | 219 | 660 |
| | $\log_2 b$ | 11 | - | - | 11 | - | - | 11 | - | - |

# Appendix B  Fast tensor power method via symmetric sketching

In this section we show how to do fast tensor power method using symmetric tensor sketches. More specifically, we explain how to approximately compute $\mathbf{T}(\boldsymbol{u}, \boldsymbol{u}, \boldsymbol{u})$ and $\mathbf{T}(\mathbf{I}, \boldsymbol{u}, \boldsymbol{u})$ when colliding hashes are used.

For symmetric tensors $\mathbf{A}$ and $\mathbf{B}$, their inner product can be approximated by

$$\langle \mathbf{A}, \mathbf{B} \rangle \approx \langle \tilde{s}_{\mathbf{A}}, \tilde{s}_{\widetilde{\mathbf{B}}} \rangle, \tag{10}$$

where $\widetilde{\mathbf{B}}$ is an "upper-triangular" tensor defined as

$$\widetilde{\mathbf{B}}_{i,j,k} = \begin{cases} \mathbf{B}_{i,j,k}, & \text{if } i \le j \le k; \\ 0, & \text{otherwise.} \end{cases} \tag{11}$$

Note that in Eq. (10) only the matrix $\mathbf{B}$ is "truncated". We show this gives consistent estimates of $\langle \mathbf{A}, \mathbf{B} \rangle$ in Appendix E.2.

Recall that $\mathbf{T}(\boldsymbol{u}, \boldsymbol{u}, \boldsymbol{u}) = \langle \mathbf{T}, \mathbf{X} \rangle$ where $\mathbf{X} = \boldsymbol{u} \otimes \boldsymbol{u} \otimes \boldsymbol{u}$. The symmetric tensor sketch $\tilde{s}_{\widetilde{\mathbf{X}}}$ can be computed as

$$\tilde{s}_{\widetilde{\mathbf{X}}} = \frac{1}{6} \tilde{s}_{\boldsymbol{u}}^{\otimes 3} + \frac{1}{2} \tilde{s}_{2,\boldsymbol{u} \circ \boldsymbol{u}} * \tilde{s}_{\boldsymbol{u}} + \frac{1}{3} \tilde{s}_{3,\boldsymbol{u} \circ \boldsymbol{u} \circ \boldsymbol{u}}, \tag{12}$$

where $\tilde{s}_{2,\boldsymbol{u} \circ \boldsymbol{u}}(t) = \sum_{2h(i)=t} \sigma(i)^2 \boldsymbol{u}_i^2$ and $\tilde{s}_{3,\boldsymbol{u} \circ \boldsymbol{u} \circ \boldsymbol{u}}(t) = \sum_{3h(i)=t} \sigma(i)^3 \boldsymbol{u}_i^3$. As a result,

$$\mathbf{T}(\boldsymbol{u}, \boldsymbol{u}, \boldsymbol{u}) \approx \frac{1}{6} \langle \mathcal{F}(\tilde{s}_{\mathbf{T}}), \mathcal{F}(\tilde{s}_{\boldsymbol{u}}) \circ \mathcal{F}(\tilde{s}_{\boldsymbol{u}}) \circ \mathcal{F}(\tilde{s}_{\boldsymbol{u}}) \rangle + \frac{1}{2} \langle \mathcal{F}(\tilde{s}_{\mathbf{T}}), \mathcal{F}(\tilde{s}_{2,\boldsymbol{u} \circ \boldsymbol{u}}) \circ \mathcal{F}(\tilde{s}_{\boldsymbol{u}}) \rangle + \frac{1}{3} \langle \tilde{s}_{\mathbf{T}}, \tilde{s}_{3,\boldsymbol{u} \circ \boldsymbol{u} \circ \boldsymbol{u}} \rangle. \tag{13}$$

---

**Algorithm 2** Fast ALS method

---

1: **Input**: $\mathbf{T} \in \mathbb{R}^{n \times n \times n}$, target rank $k$, $T$, $B$, $b$.

2: **Initialize**: $B$ independent index hash functions $h^{(1)}, \cdots, h^{(B)}$ and $\sigma^{(1)}, \cdots, \sigma^{(B)}$; random matrices $\mathbf{A}, \mathbf{B}, \mathbf{C} \in \mathbb{R}^{n \times k}$; $\{\lambda_i\}_{i=1}^k$.

3: For $m = 1, \cdots, B$ compute $\boldsymbol{s}_{\mathbf{T}}^{(m)} \in \mathbb{C}^b$.

4: **for** $t = 1$ to $T$ **do**

5:     Compute count sketches $\boldsymbol{s}_{\boldsymbol{b}_i}$, $\boldsymbol{s}_{\boldsymbol{c}_i}$ for $i = 1, \cdots, k$. For each $i = 1, \cdots, k; m = 1, \cdots, b$ compute $\boldsymbol{v}_i^{(m)} \approx \mathbf{T}(\mathbf{I}, \boldsymbol{b}_i, \boldsymbol{c}_i)$.

6:     $\bar{\boldsymbol{v}}_{ij} \leftarrow \text{med}(\Re(\boldsymbol{v}_{ij}^{(1)}), \Re(\boldsymbol{v}_{ij}^{(2)}), \cdots, \Re(\boldsymbol{v}_{ij}^{(B)}))$.

7:     Set $\widehat{\mathbf{A}} = \{\bar{\boldsymbol{v}}\}_{ij}$ and $\hat{\lambda}_i = \|\hat{\boldsymbol{a}}_i\|$; afterwards, normalize each column of $\mathbf{A}$.

8:     Update $\mathbf{B}$ and $\mathbf{C}$ similarly.

9: **Output**: eigenvalues $\{\lambda_i\}_{i=1}^k$; solutions $\mathbf{A}, \mathbf{B}, \mathbf{C}$.

---

For $\mathbf{T}(\mathbf{I}, \boldsymbol{u}, \boldsymbol{u})$ recall that $[\mathbf{T}(\mathbf{I}, \boldsymbol{u}, \boldsymbol{u})]_i = \langle \mathbf{T}, \mathbf{Y}_i \rangle$ where $\mathbf{Y}_i = \boldsymbol{e}_i \otimes \boldsymbol{u} \otimes \boldsymbol{u}$. We first symmetrize it by defining $\mathbf{Z}_i = \boldsymbol{e}_i \otimes \boldsymbol{u} \otimes \boldsymbol{u} + \boldsymbol{u} \otimes \boldsymbol{e}_i \otimes \boldsymbol{u} + \boldsymbol{u} \otimes \boldsymbol{u} \otimes \boldsymbol{e}_i$. [5] The sketch of $\widetilde{\mathbf{Z}}_i$ can be subsequently computed as

$$\tilde{\boldsymbol{s}}_{\widetilde{\mathbf{Z}}_i} = \frac{1}{2} \tilde{\boldsymbol{s}}_{\boldsymbol{u}} * \tilde{\boldsymbol{s}}_{\boldsymbol{u}} * \tilde{\boldsymbol{s}}_{\boldsymbol{e}_i} + \frac{1}{2} \tilde{\boldsymbol{s}}_{2, \boldsymbol{u} \circ \boldsymbol{u}} * \tilde{\boldsymbol{s}}_{\boldsymbol{e}_i} + \tilde{\boldsymbol{s}}_{2, \boldsymbol{e}_i \circ \boldsymbol{u}} * \tilde{\boldsymbol{s}}_{\boldsymbol{u}} + \tilde{\boldsymbol{s}}_{3, \boldsymbol{e}_i \circ \boldsymbol{u} \circ \boldsymbol{u}}. \tag{14}$$

Consequently,

$$\mathbf{T}(\mathbf{I}, \boldsymbol{u}, \boldsymbol{u}) \approx \left\langle \mathcal{F}^{-1}\left( \mathcal{F}(\tilde{\boldsymbol{s}}_{\mathbf{T}}) \circ \overline{\mathcal{F}(\tilde{\boldsymbol{s}}_{\boldsymbol{u}})} \right), \tilde{\boldsymbol{s}}_{2, \boldsymbol{e}_i \circ \boldsymbol{u}} \right\rangle + \frac{1}{6} \left\langle \mathcal{F}^{-1}\left( \mathcal{F}(\tilde{\boldsymbol{s}}_{\mathbf{T}}) \circ \overline{\mathcal{F}(\tilde{\boldsymbol{s}}_{\boldsymbol{u}})} \circ \overline{\mathcal{F}(\tilde{\boldsymbol{s}}_{\boldsymbol{u}})} \right), \tilde{\boldsymbol{s}}_{\boldsymbol{e}_i} \right\rangle$$
$$+ \frac{1}{6} \left\langle \mathcal{F}^{-1}\left( \mathcal{F}(\tilde{\boldsymbol{s}}_{\mathbf{T}}) \circ \overline{\mathcal{F}(\tilde{\boldsymbol{s}}_{2, \boldsymbol{u} \circ \boldsymbol{u}})} \right), \tilde{\boldsymbol{s}}_{\boldsymbol{e}_i} \right\rangle + \langle \tilde{\boldsymbol{s}}_{\mathbf{T}}, \tilde{\boldsymbol{s}}_{3, \boldsymbol{e}_i \circ \boldsymbol{u} \circ \boldsymbol{u}} \rangle. \tag{15}$$

Note that all of $\tilde{\boldsymbol{s}}_{\boldsymbol{e}_i}$, $\tilde{\boldsymbol{s}}_{2, \boldsymbol{e}_i \circ \boldsymbol{u}}$ and $\tilde{\boldsymbol{s}}_{3, \boldsymbol{e}_i \circ \boldsymbol{u} \circ \boldsymbol{u}}$ have exactly one nonzero entries. So we can pre-compute all terms on the left sides of inner products in Eq. (15) and then read off the values for each entry in $\mathbf{T}(\mathbf{I}, \boldsymbol{u}, \boldsymbol{u})$.

## Appendix C    Fast ALS: method and simulation result

In this section we describe how to use tensor sketching to accelerate the Alternating Least Squares (ALS) method for tensor CP decomposition. We also provide experimental results on synthetic data and compare our fast ALS implementation with the Matlab tensor toolbox [32, 33], which is widely considered to be the state-of-the-art for tensor decomposition.

### C.1   Alternating Least Squares

Alternating Least Squares (ALS) is a popular method for tensor CP decompositions [19]. The algorithm maintains $\boldsymbol{\lambda} \in \mathbb{R}^k$, $\mathbf{A}, \mathbf{B}, \mathbf{C} \in \mathbb{R}^{n \times k}$ and iteratively perform the following update steps:

$$\widehat{\mathbf{A}} = \mathbf{T}_{(1)}(\mathbf{C} \odot \mathbf{B})(\mathbf{C}^\top \mathbf{C} \circ \mathbf{B}^\top \mathbf{B})^\dagger. \tag{16}$$
$$\widehat{\mathbf{B}} = \mathbf{T}_{(1)}(\widehat{\mathbf{A}} \odot \mathbf{C})(\widehat{\mathbf{A}}^\top \widehat{\mathbf{A}} \circ \mathbf{C}^\top \mathbf{C})^\dagger;$$
$$\widehat{\mathbf{C}} = \mathbf{T}_{(1)}(\widehat{\mathbf{B}} \odot \widehat{\mathbf{A}})(\widehat{\mathbf{B}}^\top \widehat{\mathbf{B}} \circ \widehat{\mathbf{A}}^\top \widehat{\mathbf{A}})^\dagger.$$

After each update, $\hat{\lambda}_r$ is set to $\|\boldsymbol{a}_r\|_2$ (or $\|\boldsymbol{b}_r\|_2$, $\|\boldsymbol{c}_r\|_2$) for $r = 1, \cdots, k$ and the matrix $\mathbf{A}$ (or $\mathbf{B}$, $\mathbf{C}$) is normalized so that each column has unit norm. The final low-rank approximation is obtained by $\sum_{i=1}^k \hat{\lambda}_i \hat{\boldsymbol{a}}_i \otimes \hat{\boldsymbol{b}}_i \otimes \hat{\boldsymbol{c}}_i$.

There is no guarantee that ALS converges or gives a good tensor decomposition. Nevertheless, it works reasonably well in most applications [19]. In general ALS requires $O(T(n^3 k + k^3))$ computations and $O(n^3)$ storage, where $T$ is the number of iterations.

Table 7: Squared residual norm on top 10 recovered eigenvectors of 1000d tensors and running time (excluding I/O and sketch building time) for plain (exact) and sketched ALS algorithms. Two vectors are considered mismatched (wrong) if $\|\boldsymbol{v} - \hat{\boldsymbol{v}}\|_2^2 > 0.1$.

| | | Residual norm | | | | | No. of wrong vectors | | | | | Running time (min.) | | | | |
|---|---|---|---|---|---|---|---|---|---|---|---|---|---|---|---|---|
| | $\log_2(b)$: | 12 | 13 | 14 | 15 | 16 | 12 | 13 | 14 | 15 | 16 | 12 | 13 | 14 | 15 | 16 |
| $\sigma = .01$ | $B = 20$ | .71 | .41 | .25 | .17 | .12 | 10 | 9 | 7 | 6 | 4 | .11 | .22 | .49 | 1.1 | 2.4 |
| | $B = 30$ | .50 | .34 | .21 | .14 | .11 | 9 | 8 | 7 | 5 | 3 | .17 | .33 | .75 | 1.6 | 3.5 |
| | $B = 40$ | .46 | .28 | .17 | .10 | **.07** | 9 | 8 | 6 | 5 | **1** | .23 | .45 | 1.0 | 2.2 | **4.7** |
| | Exact† | .07 | | | | | 1 | | | | | 22.8 | | | | |
| $\sigma = .1$ | $B = 20$ | .88 | .50 | .35 | .28 | .23 | 10 | 8 | 7 | 6 | 6 | .13 | .32 | .78 | 1.5 | 3.2 |
| | $B = 30$ | .78 | .44 | .30 | .24 | .21 | 9 | 8 | 7 | 5 | 6 | .21 | .50 | 1.1 | 2.2 | 4.7 |
| | $B = 40$ | .56 | .38 | .28 | .19 | **.16** | 9 | 8 | 6 | 4 | **2** | .29 | .69 | 1.5 | 3.5 | **6.3** |
| | Exact† | .17 | | | | | 2 | | | | | 32.3 | | | | |

†Calling `cp_als` in Matlab tensor toolbox. It is run for exactly $T = 30$ iterations.

## C.2 Accelerated ALS via sketching

Similar to robust tensor power method, the ALS algorithm can be significantly accelerated by using the idea of sketching as shown in this work. However, for ALS we cannot use colliding hashes because though the input tensor $\mathbf{T}$ is symmetric, its CP decomposition is not since we maintain three different solution matrices $\mathbf{A}$, $\mathbf{B}$ and $\mathbf{C}$. As a result, we roll back to asymmetric tensor sketches defined in Eq. (1). Recall that given $\mathbf{A}, \mathbf{B}, \mathbf{C} \in \mathbb{R}^{n \times k}$ we want to compute

$$\hat{\mathbf{A}} = \mathbf{T}_{(1)}(\mathbf{C} \odot \mathbf{B})(\mathbf{C}^\top \mathbf{C} \circ \mathbf{B}^\top \mathbf{B})^\dagger. \tag{17}$$

When $k$ is much smaller than the ambient tensor dimension $n$ the computational bottleneck of Eq. (17) is $\mathbf{T}_{(1)}(\mathbf{C} \odot \mathbf{B})$, which requires $O(n^3 k)$ operations. Below we show how to use sketching to speed up this computation.

Let $\boldsymbol{x} \in \mathbb{R}^{n^2}$ be one row in $\mathbf{T}_{(1)}$ and consider $(\mathbf{C} \odot \mathbf{B})^\top \boldsymbol{x}$. It can be shown that [15]

$$\left[ (\mathbf{C} \odot \mathbf{B})^\top \boldsymbol{x} \right]_i = \boldsymbol{b}_i^\top \mathbf{X} \boldsymbol{c}_i, \quad \forall i = 1, \cdots, k, \tag{18}$$

where $\mathbf{X} \in \mathbb{R}^{n \times n}$ is the reshape of vector $\boldsymbol{x}$. Subsequently, the product $\mathbf{T}_{(1)}(\mathbf{C} \odot \mathbf{B})$ can be re-written as

$$\mathbf{T}_{(1)}(\mathbf{C} \odot \mathbf{B}) = [\mathbf{T}(\mathbf{I}, \boldsymbol{b}_1, \boldsymbol{c}_1); \cdots ; \mathbf{T}(\mathbf{I}, \boldsymbol{b}_k, \boldsymbol{c}_k)]. \tag{19}$$

Using Proposition 1 we can compute each of $\mathbf{T}(\mathbf{I}, \boldsymbol{b}_i, \boldsymbol{c}_i)$ in $O(n + b \log b)$ iterations. Note that in general $\boldsymbol{b}_i \neq \boldsymbol{c}_i$, but Proposition 1 still holds by replacing one of the two $\boldsymbol{s_u}$ sketches. As a result, $\mathbf{T}_{(1)}(\mathbf{C} \odot \mathbf{B})$ can be computed in $O(k(n + b \log b))$ operations once $\boldsymbol{s_T}$ is computed. The pseudocode of fast ALS is listed in Algorithm 2. Its time complexity and space complexity are $O(T(k(n + Bb \log b) + k^3))$ (excluding the time for building $\boldsymbol{s_T}$) and $O(Bb)$, respectively.

## C.3 Simulation results

We compare the performance of fast ALS with a brute-force implementation under various hash length settings on synthetic datasets in Table 7. Settings for generating the synthetic dataset is exactly the same as in Section 5.1. We use the `cp_als` routine in Matlab tensor toolbox as the reference brute-force implementation of ALS. For fair comparison, exactly $T = 30$ iterations are performed for both plain and accelerated ALS algorithms. Table 7 shows that when sketch length $b$ is not too small, fast ALS achieves comparable accuracy with exact methods while being much faster in terms of running time.

## Appendix D  Spectral LDA and fast spectral LDA

Latent Dirichlet Allocation (LDA, [3]) is a powerful tool in topic modeling. In this section we first review the LDA model and introduce the tensor decomposition method for learning LDA models, which was proposed in [1]. We then provide full details of our proposed fast spectral LDA algorithm. Pseudocode for fast spectral LDA is listed in Algorithm 3.

**Algorithm 3** Fast spectral LDA

1: **Input**: Unlabeled documents, $V$, $K$, $\alpha_0$, $B$, $b$.
2: Compute empirical moments $\widehat{\mathbf{M}}_1$ and $\widehat{\mathbf{M}}_2$ defined in Eq. (20,21).
3: $[\mathbf{U}, \mathbf{S}, \mathbf{V}] \leftarrow \text{truncatedSVD}(\widehat{\mathbf{M}}_2, k)$; $\mathbf{W}_{ik} \leftarrow \frac{\mathbf{U}_{ik}}{\sqrt{\sigma_k}}$.
4: Build $B$ tensor sketches of $\widehat{\mathbf{M}}_3(\mathbf{W}, \mathbf{W}, \mathbf{W})$.
5: Find CP decomposition $\{\lambda_i\}_{i=1}^k$, $\mathbf{A} = \mathbf{B} = \mathbf{C} = \{\boldsymbol{v}_i\}_{i=1}^k$ of $\widehat{\mathbf{M}}_3(\mathbf{W}, \mathbf{W}, \mathbf{W})$ using either fast tensor power method or fast ALS method.
6: **Output**: estimates of prior parameters $\hat{\alpha}_i = \frac{4\alpha_0(\alpha_0+1)}{(\alpha_0+2)^2\lambda_i^2}$ and topic distributions $\hat{\boldsymbol{\mu}}_i = \frac{\alpha_0+2}{2}\lambda_i(\mathbf{W}^\dagger)^\top\boldsymbol{v}_i$.

---

## D.1 LDA and spectral LDA

LDA models a collection of documents by a topic dictionary $\boldsymbol{\Phi} \in \mathbb{R}^{V \times K}$ and a Dirichlet prior $\boldsymbol{\alpha} \in \mathbb{R}^k$, where $V$ is the vocabulary size and $k$ is the number of topics. Each column in $\boldsymbol{\Phi}$ is a probability distribution (i.e., non-negative and sum to one) representing the word distribution of a particular topic. For each document $d$, a topic mixing vector $\boldsymbol{h}_d \in \mathbb{R}^k$ is first sampled from a Dirichlet distribution parameterized by $\boldsymbol{\alpha}$. Afterwards, words in document $d$ i.i.d. sampled from a categorical distribution parameterized by $\boldsymbol{\Phi}\boldsymbol{h}_d$.

A spectral method for LDA based on 3rd-order robust tensor decomposition was proposed in [1] to provably learn LDA model parameters from a polynomial number of training documents. Let $\boldsymbol{x} \in \mathbb{R}^V$ represent a single word; that is, for word $w$ we have $x_w = 1$ and $x_{w'} = 0$ for all $w' \neq w$. Define first, second and third order moments $\mathbf{M}_1, \mathbf{M}_2$ and $\mathbf{M}_3$ as follows:

$$\mathbf{M}_1 = \mathbb{E}[\boldsymbol{x}_1]; \tag{20}$$

$$\mathbf{M}_2 = \mathbb{E}[\boldsymbol{x}_1 \otimes \boldsymbol{x}_2] - \frac{\alpha_0}{\alpha_0 + 1}\mathbf{M}_1 \otimes \mathbf{M}_1; \tag{21}$$

$$\mathbf{M}_3 = \mathbb{E}[\boldsymbol{x}_1 \otimes \boldsymbol{x}_2 \otimes \boldsymbol{x}_3] - \frac{\alpha_0}{\alpha_0 + 2}(\mathbb{E}[\boldsymbol{x}_1 \otimes \boldsymbol{x}_2 \otimes \mathbf{M}_1] + \mathbb{E}[\boldsymbol{x}_1 \otimes \mathbf{M}_1 \otimes \boldsymbol{x}_2] + \mathbb{E}[\mathbf{M}_1 \otimes \boldsymbol{x}_1 \otimes \boldsymbol{x}_2])$$

$$+ \frac{2\alpha_0^2}{(\alpha_0 + 1)(\alpha_0 + 2)}\mathbf{M}_1 \otimes \mathbf{M}_1 \otimes \mathbf{M}_1. \tag{22}$$

Here $\alpha_0 = \sum_k \alpha_k$ is assumed to be a known quantity. Using elementary algebra it can be shown that

$$\mathbf{M}_2 = \frac{1}{\alpha_0(\alpha_0 + 1)}\sum_{i=1}^k \alpha_i\boldsymbol{\mu}_i\boldsymbol{\mu}_i^\top; \tag{23}$$

$$\mathbf{M}_3 = \frac{2}{\alpha_0(\alpha_0 + 1)(\alpha_0 + 2)}\sum_{i=1}^k \alpha_i\boldsymbol{\mu}_i \otimes \boldsymbol{\mu}_i \otimes \boldsymbol{\mu}_i. \tag{24}$$

To extract topic vectors $\{\boldsymbol{\mu}_i\}_{i=1}^k$ from $\mathbf{M}_2$ and $\mathbf{M}_3$, a *simultaneous diagonalization* procedure is carried out. More specifically, the algorithm first finds a whitening matrix $\mathbf{W} \in \mathbb{R}^{V \times K}$ with orthonormal columns such that $\mathbf{W}^\top\mathbf{M}_2\mathbf{W} = \mathbf{I}_{K \times K}$. In practice, this step can be completed by performing a truncated SVD on $\mathbf{M}_2$, $\mathbf{M}_2 = \mathbf{U}_K\boldsymbol{\Sigma}_K\mathbf{V}_K$, and set $\mathbf{W}_{ik} = \mathbf{U}_{ik}/\sqrt{\boldsymbol{\Sigma}_{kk}}$. Afterwards, tensor CP decomposition is performed on the whitened third order moment $\mathbf{M}_3(\mathbf{W}, \mathbf{W}, \mathbf{W})$ [6] to obtain a set of eigenvectors $\{\boldsymbol{v}_k\}_{k=1}^K$. The topic vectors $\{\boldsymbol{\mu}_k\}_{k=1}^K$ can be subsequently obtained by multiplying $\{\boldsymbol{v}_k\}_{k=1}^K$ with the pseudoinverse of $\mathbf{W}$. Note that Eq. (20,21,22) are defined in exact word moments. In practice we use empirical moments (e.g., word frequency vector and co-occurrence matrix) to approximate these exact moments.

## D.2 Fast spectral LDA

To further accelerate the spectral method mentioned in the previous section, it helps to first identify computational bottlenecks of spectral LDA. In general, the computation of $\widehat{\mathbf{M}}_1, \widehat{\mathbf{M}}_2$ and the whitening step are not the computational bottleneck when $V$ is not too large and each document is not too long. The bottleneck comes from the computation of (the sketch of) $\widehat{\mathbf{M}}_3(\mathbf{W}, \mathbf{W}, \mathbf{W})$ and its tensor decomposition. By Eq. (22), the computation of $\widehat{\mathbf{M}}_3(\mathbf{W}, \mathbf{W}, \mathbf{W})$ reduces to computing $\widehat{\mathbf{M}}_1^{\otimes 3}(\mathbf{W}, \mathbf{W}, \mathbf{W})$, $\hat{\mathbb{E}}[\boldsymbol{x}_1 \otimes \boldsymbol{x}_2 \otimes \widehat{\mathbf{M}}_1](\mathbf{W}, \mathbf{W}, \mathbf{W})$, [7] and $\hat{\mathbb{E}}[\boldsymbol{x}_1 \otimes \boldsymbol{x}_2 \otimes \boldsymbol{x}_3](\mathbf{W}, \mathbf{W}, \mathbf{W})$. The first term $\widehat{\mathbf{M}}_1^{\otimes 3}(\mathbf{W}, \mathbf{W}, \mathbf{W})$ poses no particular challenge as it can be written as $(\mathbf{W}^\top \widehat{\mathbf{M}}_1)^{\otimes 3}$. Its sketch can then be efficiently obtained by applying techniques in Section 3.1. In the remainder of this section we focus on efficient computation of the sketch of the other two terms mentioned above.

We first show how to efficiently sketching $\hat{\mathbb{E}}[\boldsymbol{x}_1 \otimes \boldsymbol{x}_2 \otimes \boldsymbol{x}_3](\mathbf{W}, \mathbf{W}, \mathbf{W})$ given the whitening matrix $\mathbf{W}$ and $D$ training documents. Let $\mathbf{T}\hat{\mathbb{E}}[\boldsymbol{x}_1 \otimes \boldsymbol{x}_2 \otimes \boldsymbol{x}_3](\mathbf{W}, \mathbf{W}, \mathbf{W})$ denote the whitened $k \times k \times k$ tensor to be sketched and write $\mathbf{T} = \sum_{d=1}^D \mathbf{T}_d$, where $\mathbf{T}_d$ is the contribution of the $d$th training document to $\mathbf{T}$. By definition, $\mathbf{T}_d$ can be expressed as $\mathbf{T}_d = \mathbf{N}_d(\mathbf{W}, \mathbf{W}, \mathbf{W})$, where $\mathbf{W}$ is the $V \times k$ whitening matrix and $\mathbf{N}_d$ is the $V \times V \times V$ empirical moment tensor computed on the $d$th document. More specifically, for $i, j, k \in \{1, \cdots, V\}$ we have

$$\mathbf{N}_{d,ijk} = \frac{1}{m_d(m_d - 1)(m_d - 2)} \begin{cases} n_{di}(n_{dj} - 1)(n_{dk} - 2), & i = j = k; \\ n_{di}(n_{di} - 1)n_{dk}, & i = j, j \neq k; \\ n_{di}n_{dj}(n_{dj} - 1) & j = k, i \neq j; \\ n_{di}(n_{di} - 1)n_{dj}, & i = k, i \neq j; \\ n_{di}n_{dj}n_{dk}, & \text{otherwise.} \end{cases}$$

Here $m_d$ is the length (i.e., number of words) of document $d$ and $\boldsymbol{n}_d \in \mathbb{R}^V$ is the corresponding word count vector. Previous straightforward implementation require at least $O(k^3 + m_d k^2)$ operations per document to build the tensor $\mathbf{T}$ and $O(k^4 LT)$ to decompose it [30, 29], which is prohibitively slow for real-world applications. In section 3 we discussed how to decompose a tensor efficiently once we have its sketch. We now show how to build the sketch of $\mathbf{T}$ efficiently from document word counts $\{\boldsymbol{n}_d\}_{d=1}^D$.

By definition, $\mathbf{T}_d$ can be decomposed as

$$\mathbf{T}_d = \boldsymbol{p}^{\otimes 3} - \sum_{i=1}^V n_i (\boldsymbol{w}_i \otimes \boldsymbol{w}_i \otimes \boldsymbol{p} + \boldsymbol{w}_i \otimes \boldsymbol{p} \otimes \boldsymbol{w}_i + \boldsymbol{p} \otimes \boldsymbol{w}_i \otimes \boldsymbol{w}_i) + \sum_{i=1}^V 2n_i \boldsymbol{w}_i^{\otimes 3}, \qquad (25)$$

where $\boldsymbol{p} = \mathbf{W}\boldsymbol{n}$ and $\boldsymbol{w}_i \in \mathbb{R}^k$ is the $i$th row of the whitening matrix $\mathbf{W}$. A direct implementation is to sketch each of the low-rank components in Eq. (25) and compute their sum. Since there are $O(m_d)$ tensors, building the sketch of $\mathbf{T}_d$ requires $O(m_d)$ FFTs, which is unsatisfactory. However, note that $\{\boldsymbol{w}_i\}_{i=1}^V$ are fixed and shared across documents. So when scanning the documents we maintain the sum of $n_i$ and $n_i \boldsymbol{p}$ and add the incremental after all documents are scanned. In this way, we only need $O(1)$ FFT per document with an additional $O(V)$ FFTs. Since the total number of documents $D$ is usually much larger than $V$, this provides significant speed-ups over the naive method that sketches each term in Eq. (25) independently. As a result, the sketch of $\mathbf{T}$ can be computed in $O(k(\sum_d m_d) + (D + V)b \log b)$ operations, which is much more efficient than the $O(k^2(\sum_d m_d) + Dk^3)$ brute-force computation.

We next turn to the term $\hat{\mathbb{E}}[\boldsymbol{x}_1 \otimes \boldsymbol{x}_2 \otimes \widehat{\mathbf{M}}_1](\mathbf{W}, \mathbf{W}, \mathbf{W})$. Fix a document $d$ and let $\boldsymbol{p} = \mathbf{W}\boldsymbol{n}_d$. Define $\boldsymbol{q} = \mathbf{W}\widehat{\mathbf{M}}_1$. By definition, the whitened empirical moment can be decomposed as

$$\hat{\mathbb{E}}[\boldsymbol{x}_1 \otimes \boldsymbol{x}_2 \otimes \widehat{\mathbf{M}}_1](\mathbf{W}, \mathbf{W}, \mathbf{W}) = \sum_{i=1}^V n_i \boldsymbol{p} \otimes \boldsymbol{p} \otimes \boldsymbol{q}, \qquad (26)$$

Note that Eq. (26) is very similar to Eq. (25). Consequently, we can apply the same trick (i.e., adding $\boldsymbol{p}$ and $n_i \boldsymbol{p}$ up before doing sketching or FFT) to compute Eq. (26) efficiently.

## Appendix E  Proofs

### E.1  Proofs of some technical propositions

*Proof of Proposition 2.*  We prove the proposition for the case $q = 2$ (i.e., $\tilde{H}$ is 2-wise independent). This suffices for our purpose in this paper and generalization to $q > 2$ cases is straightforward. For notational simplicity we omit all modulo operators. Consider two $p$-tuples $\boldsymbol{l} = (l_1, \cdots, l_p)$ and $\boldsymbol{l}' = (l'_1, \cdots, l'_p)$ such that $\boldsymbol{l} \neq \boldsymbol{l}'$. Since $\tilde{H}$ is permutation invariant, we assume without loss of generality that for some $s < p$ and $1 \leq i \leq s$ we have $l_i = l'_i$. Fix $t, t' \in [b]$. We then have

$$\Pr[\tilde{H}(\boldsymbol{l}) = t \wedge \tilde{H}(\boldsymbol{l}') = t'] = \sum_{a} \sum_{h(l_1) + \cdots + h(l_s) = a} \Pr[h(l_1) + \cdots + h(l_s) = a]$$

$$\cdot \sum_{\substack{r_{s+1} + \cdots + r_p = t-a \\ r'_{s+1} + \cdots + r'_p = t'-a}} \Pr[h(l_{s+1}) = r_1 \wedge \cdots \wedge h(l_p) = r_p \wedge h(l'_{s+1}) = r'_1 \wedge \cdots \wedge h(l'_p) = r'_p]. \quad (27)$$

Since $h$ is $2p$-wise independent, we have

$$\Pr[h(l_1) + \cdots + h(l_s) = a] = \sum_{r_1 + \cdots + r_s = a} \Pr[h(l_1) = r_1 \wedge \cdots h(l_s) = r_s] = b^{s-1} \cdot \frac{1}{b^s} = \frac{1}{b};$$

$$\sum_{\substack{r_{s+1} + \cdots + r_p = t-a \\ r'_{s+1} + \cdots + r'_p = t-a}} \Pr[h(l_{s+1}) = r_1 \wedge \cdots \wedge h(l_p) = r_p \wedge h(l'_{s+1}) = r'_1 \wedge \cdots \wedge h(l'_p) = r'_p]$$

$$= b^{2(p-s-1)} \cdot \frac{1}{b^{2(p-s)}} = \frac{1}{b^2}.$$

Summing everything up we get $\Pr[\tilde{H}(\boldsymbol{l}) = t \wedge \tilde{H}(\boldsymbol{l}') = t'] = 1/b^2$, which is to be demonstrated. $\quad\square$

*Proof of Proposition 1.*  Since both FFT and inverse FFT preserve inner products, we have

$$\langle \boldsymbol{s_T}, \boldsymbol{s_{1,u}} * \boldsymbol{s_{2,u}} * \boldsymbol{s_{3,e_i}} \rangle = \langle \mathcal{F}(\boldsymbol{s_T}), \mathcal{F}(\boldsymbol{s_{1,u}}) \circ \mathcal{F}(\boldsymbol{s_{2,u}}) \circ \mathcal{F}(\boldsymbol{s_{3,e_i}}) \rangle$$

$$= \langle \mathcal{F}(\boldsymbol{s_T}) \circ \overline{\mathcal{F}(\boldsymbol{s_{1,u}})} \circ \overline{\mathcal{F}(\boldsymbol{s_{2,u}})}, \mathcal{F}(\boldsymbol{s_{3,e_i}}) \rangle$$

$$= \langle \mathcal{F}^{-1}(\mathcal{F}(\boldsymbol{s_T}) \circ \overline{\mathcal{F}(\boldsymbol{s_{1,u}})} \circ \overline{\mathcal{F}(\boldsymbol{s_{2,u}})}), \boldsymbol{s_{3,e_i}} \rangle.$$

$$\square$$

### E.2  Analysis of tensor sketch approximation error

Proofs of Theorem 1 is based on the following two key lemmas, which states that $\langle \tilde{\boldsymbol{s}}_\mathbf{A}, \tilde{\boldsymbol{s}}_{\widetilde{\mathbf{B}}} \rangle$ is a consistent estimator of the true inner product $\langle \mathbf{A}, \mathbf{B} \rangle$; furthermore, the variance of the estimator decays linearly with the hash length $b$. The lemmas are interesting in their own right, providing useful tools for proving approximation accuracy in a wide range of applications when colliding hash and symmetric sketches are used.

**Lemma 1.** *Suppose $\mathbf{A}, \mathbf{B} \in \bigotimes^p \mathbb{R}^n$ are two symmetric real tensors and let $\tilde{\boldsymbol{s}}_\mathbf{A}, \tilde{\boldsymbol{s}}_{\widetilde{\mathbf{B}}} \in \mathbb{C}^b$ be the symmetric tensor sketches of $\mathbf{A}$ and $\widetilde{\mathbf{B}}$. That is,*

$$\tilde{s}_\mathbf{A}(t) = \sum_{\tilde{H}(i_1, \cdots, i_p) = t} \sigma_{i_1} \cdots \sigma_{i_p} \mathbf{A}_{i_1, \cdots, i_p}; \quad (28)$$

$$\tilde{s}_{\widetilde{\mathbf{B}}}(t) = \sum_{\substack{\tilde{H}(i_1, \cdots, i_p) = t \\ i_1 \leq \cdots \leq i_p}} \sigma_{i_1} \cdots \sigma_{i_p} \mathbf{B}_{i_1, \cdots, i_p}. \quad (29)$$

*Assume $\tilde{H}(i_1, \cdots, i_p) = (h(i_1) + \cdots + h(i_p)) \mod b$ are drawn from a 2-wise independent hash family. Then the following holds:*

$$\mathbb{E}_{h,\sigma} \left[ \langle \tilde{\boldsymbol{s}}_\mathbf{A}, \tilde{\boldsymbol{s}}_{\widetilde{\mathbf{B}}} \rangle \right] = \langle \mathbf{A}, \mathbf{B} \rangle, \quad (30)$$

$$\mathbb{V}_{h,\sigma} \left[ \langle \tilde{\boldsymbol{s}}_\mathbf{A}, \tilde{\boldsymbol{s}}_{\widetilde{\mathbf{B}}} \rangle \right] \leq \frac{4^p \|\mathbf{A}\|_F^2 \|\mathbf{B}\|_F^2}{b}. \quad (31)$$

**Lemma 2.** *Following notations and assumptions in Lemma 1. Let $\{\mathbf{A}_i\}_{i=1}^m$ and $\{\mathbf{B}_i\}_{i=1}^m$ be symmetric real $n \times n \times n$ tensors and fix real vector $\boldsymbol{w} \in \mathbb{R}^m$. Then we have*

$$\mathbb{E}\left[\sum_{i,j} w_i w_j \langle \tilde{\boldsymbol{s}}_{\mathbf{A}_i}, \tilde{\boldsymbol{s}}_{\widetilde{\mathbf{B}}_j} \rangle \right] = \sum_{i,j} w_i w_j \langle \mathbf{A}_i, \mathbf{B}_j \rangle; \tag{32}$$

$$\mathbb{V}\left[\sum_{i,j} w_i w_j \langle \tilde{\boldsymbol{s}}_{\mathbf{A}_i}, \tilde{\boldsymbol{s}}_{\widetilde{\mathbf{B}}_j} \rangle \right] \leq \frac{4^p \|\boldsymbol{w}\|^4 (\max_i \|\mathbf{A}_i\|_F^2)(\max_i \|\mathbf{B}_i\|_F^2)}{b}. \tag{33}$$

*Proof of Lemma 1.* We first define some notations. Let $\boldsymbol{l} = (l_1, \cdots, l_p) \in [d]^p$ be a $p$-tuple denoting a multi-index. Define $\mathbf{A}_{\boldsymbol{l}} := \mathbf{A}_{l_1, \cdots, l_p}$ and $\sigma(\boldsymbol{l}) := \sigma_{l_1} \cdots \sigma_{l_p}$. For $\boldsymbol{l}, \boldsymbol{l}' \in [n]^p$, define $\delta(\boldsymbol{l}, \boldsymbol{l}') = 1$ if $h(l_1) + \cdots + h(l_p) \equiv h(l_1') + \cdots + h(l_p') (\mod b)$ and $\delta(\boldsymbol{l}, \boldsymbol{l}') = 0$ otherwise. For a $p$-tuple $\boldsymbol{l} \in [n]^p$, let $\mathcal{L}(\boldsymbol{l}) \in [n]^p$ denote the $p$-tuple obtained by re-ordering indices in $\boldsymbol{l}$ in ascending order. Let $\mathcal{M}(\boldsymbol{l}) \in \mathbb{N}^b$ denote the "expanded version" of $\boldsymbol{l}$. That is, $[\mathcal{M}(\boldsymbol{l})]_i$ denote the number of occurrences of the index $i$ in $\boldsymbol{l}$. By definition, $\|\mathcal{M}(\boldsymbol{l})\|_1 = p$. Finally, by definition $\widetilde{\mathbf{B}}_{\boldsymbol{l}'} = \mathbf{B}_{\boldsymbol{l}'}$ if $\boldsymbol{l}' = \mathcal{L}(\boldsymbol{l}')$ and $\widetilde{\mathbf{B}}_{\boldsymbol{l}'} = 0$ otherwise.

Eq. (30) is easy to prove. By definition and linearity of expectation we have

$$\mathbb{E}[\langle \tilde{\boldsymbol{s}}_{\mathbf{A}}, \tilde{\boldsymbol{s}}_{\widetilde{\mathbf{B}}} \rangle] = \sum_{\boldsymbol{l}, \boldsymbol{l}'} \delta(\boldsymbol{l}, \boldsymbol{l}') \sigma(\boldsymbol{l}) \mathbf{A}_{\boldsymbol{l}} \bar{\sigma}(\boldsymbol{l}') \widetilde{\mathbf{B}}_{\boldsymbol{l}'}. \tag{34}$$

Note that $\delta$ and $\sigma$ are independent and

$$\mathbb{E}_\sigma[\sigma(\boldsymbol{l})\sigma(\boldsymbol{l}')] = \begin{cases} 1, & \text{if } \mathcal{L}(\boldsymbol{l}) = \mathcal{L}(\boldsymbol{l}'); \\ 0, & \text{otherwise.} \end{cases} \tag{35}$$

Also $\delta(\boldsymbol{l}, \boldsymbol{l}') = 1$ with probability 1 whenever $\mathcal{L}(\boldsymbol{l}) = \mathcal{L}(\boldsymbol{l}')$. Note that $\widetilde{\mathbf{B}}_{\boldsymbol{l}'} = 0$ whenever $\boldsymbol{l}' \neq \mathcal{L}(\boldsymbol{l}')$. Consequently,

$$\mathbb{E}[\langle \tilde{\boldsymbol{s}}_{\mathbf{A}}, \tilde{\boldsymbol{s}}_{\widetilde{\mathbf{B}}} \rangle] = \sum_{\boldsymbol{l} \in [n]^p} \mathbf{A}_{\boldsymbol{l}} \widetilde{\mathbf{B}}_{\mathcal{L}(\boldsymbol{l})} = \langle \mathbf{A}, \mathbf{B} \rangle. \tag{36}$$

For the variance, we have the following expression for $\mathbb{E}[\langle \tilde{\boldsymbol{s}}_{\mathbf{A}}, \tilde{\boldsymbol{s}}_{\widetilde{\mathbf{B}}} \rangle^2]$:

$$\mathbb{E}[\langle \tilde{\boldsymbol{s}}_{\mathbf{A}}, \tilde{\boldsymbol{s}}_{\widetilde{\mathbf{B}}} \rangle^2] = \sum_{\boldsymbol{l}, \boldsymbol{l}', \boldsymbol{r}, \boldsymbol{r}'} \mathbb{E}[\delta(\boldsymbol{l}, \boldsymbol{l}')\delta(\boldsymbol{r}, \boldsymbol{r}')] \cdot \mathbb{E}[\sigma(\boldsymbol{l})\bar{\sigma}(\boldsymbol{l}')\bar{\sigma}(\boldsymbol{r})\sigma(\boldsymbol{r}')] \cdot \mathbf{A}_{\boldsymbol{l}} \mathbf{A}_{\boldsymbol{r}} \widetilde{\mathbf{B}}_{\boldsymbol{l}'} \widetilde{\mathbf{B}}_{\boldsymbol{r}'} \quad (37)$$

$$=: \sum_{\boldsymbol{l}, \boldsymbol{l}', \boldsymbol{r}, \boldsymbol{r}'} \mathbb{E}[t(\boldsymbol{l}, \boldsymbol{l}', \boldsymbol{r}, \boldsymbol{r}')]. \tag{38}$$

We remark that $\mathbb{E}[\sigma(\boldsymbol{l})\bar{\sigma}(\boldsymbol{l}')\bar{\sigma}(\boldsymbol{r})\sigma(\boldsymbol{r}')] = 0$ if $\mathcal{M}(\boldsymbol{l}) - \mathcal{M}(\boldsymbol{l}') \neq \mathcal{M}(\boldsymbol{r}) - \mathcal{M}(\boldsymbol{r}')$. In the remainder of the proof we will assume that $\mathcal{M}(\boldsymbol{l}) - \mathcal{M}(\boldsymbol{l}') = \mathcal{M}(\boldsymbol{r}) - \mathcal{M}(\boldsymbol{r}')$. This can be further categorized into two cases:

**Case 1**: $\boldsymbol{l}' = \mathcal{L}(\boldsymbol{l})$ and $\boldsymbol{r}' = \mathcal{L}(\boldsymbol{r})$. By definition $\mathbb{E}[\sigma(\boldsymbol{l})\bar{\sigma}(\boldsymbol{l}')\sigma(\boldsymbol{r})\bar{\sigma}(\boldsymbol{r}')] = 1$ and $\mathbb{E}[\delta(\boldsymbol{l}, \boldsymbol{l}')\delta(\boldsymbol{r}, \boldsymbol{r}')] = 1$. Subsequently $\mathbb{E}[t(\boldsymbol{l}, \boldsymbol{l}', \boldsymbol{r}, \boldsymbol{r}')] = \mathbf{A}_{\boldsymbol{l}} \mathbf{A}_{\boldsymbol{r}} \widetilde{\mathbf{B}}_{\boldsymbol{l}'} \widetilde{\mathbf{B}}_{\boldsymbol{r}'}$ and hence

$$\sum_{\boldsymbol{l}, \boldsymbol{r}, \boldsymbol{l}' = \mathcal{L}(\boldsymbol{l}), \boldsymbol{r}' = \mathcal{L}(\boldsymbol{r})} \mathbb{E}[t(\boldsymbol{l}, \boldsymbol{l}', \boldsymbol{r}, \boldsymbol{r}')] = \sum_{\boldsymbol{l}, \boldsymbol{r}} \mathbf{A}_{\boldsymbol{l}} \mathbf{A}_{\boldsymbol{r}} \mathbf{B}_{\boldsymbol{l}} \mathbf{B}_{\boldsymbol{r}} = \langle \mathbf{A}, \mathbf{B} \rangle^2. \tag{39}$$

**Case 2**: $\boldsymbol{l}' \neq \mathcal{L}(\boldsymbol{l})$ or $\boldsymbol{r}' \neq \mathcal{L}(\boldsymbol{r})$. Since $\mathcal{M}(\boldsymbol{l}) - \mathcal{M}(\boldsymbol{l}') = \mathcal{M}(\boldsymbol{r}) - \mathcal{M}(\boldsymbol{r}') \neq 0$ we have $\mathbb{E}[\delta(\boldsymbol{l}, \boldsymbol{l}')\delta(\boldsymbol{r}, \boldsymbol{r}')] = 1/b$ because $h$ is a 2-wise independent hash function. In addition, $\mathbb{E}[|\sigma(\boldsymbol{l})\bar{\sigma}(\boldsymbol{l}')\sigma(\boldsymbol{r})\bar{\sigma}(\boldsymbol{r}')|] \leq 1$.

To enumerate all $(\boldsymbol{l}, \boldsymbol{l}', \boldsymbol{r}, \boldsymbol{r}')$ tuples that satisfy the colliding condition $\mathcal{M}(\boldsymbol{l}) - \mathcal{M}(\boldsymbol{l}') = \mathcal{M}(\boldsymbol{r}) - \mathcal{M}(\boldsymbol{r}') \neq 0$, we fix [8] $\|\mathcal{M}(\boldsymbol{l}) - \mathcal{M}(\boldsymbol{l}')\|_1 = 2q$ and fix $q$ positions each in $\boldsymbol{l}$ and $\boldsymbol{r}$ (for $\boldsymbol{l}'$ and $\boldsymbol{r}'$ the positions of these indices are automatically fixed because indices in $\boldsymbol{l}'$ and $\boldsymbol{r}'$ must be in ascending

order). Without loss of generality assume the fixed $q$ positions for both $\boldsymbol{l}$ and $\boldsymbol{r}$ are the first $q$ indices. The 4-tuple $(\boldsymbol{l}, \boldsymbol{r}, \boldsymbol{l}', \boldsymbol{r}')$ with $\|\mathcal{M}(\boldsymbol{l}) - \mathcal{M}(\boldsymbol{l}')\|_1 = 2q$ can then be enumerated as follows:

$$
\sum_{\substack{\boldsymbol{l}, \boldsymbol{r}, \boldsymbol{l}', \boldsymbol{r}' \\ \mathcal{M}(\boldsymbol{l}) - \mathcal{M}(\boldsymbol{l}') = \mathcal{M}(\boldsymbol{r}) - \mathcal{M}(\boldsymbol{r}') \\ \|\mathcal{M}(\boldsymbol{l}) - \mathcal{M}(\boldsymbol{l}')\|_1 = 2q}} t(\boldsymbol{l}, \boldsymbol{l}', \boldsymbol{r}, \boldsymbol{r}')
$$

$$
= \sum_{\boldsymbol{i} \in [n]^q} \sum_{\boldsymbol{j} \in [n]^q} \sum_{\substack{\boldsymbol{l} \in [n]^{p-q} \\ \boldsymbol{r} \in [n]^{p-q}}} t(\boldsymbol{i} \circ \boldsymbol{l}, \mathcal{L}(\boldsymbol{j} \circ \boldsymbol{l}), \boldsymbol{i} \circ \boldsymbol{r}, \mathcal{L}(\boldsymbol{j} \circ \boldsymbol{r}))
$$

$$
\leq \frac{1}{b} \sum_{\substack{\boldsymbol{i}, \boldsymbol{j} \in [n]^q \\ \boldsymbol{l}, \boldsymbol{r} \in [n]^{p-q}}} \mathbf{A}_{\boldsymbol{i} \circ \boldsymbol{l}} \mathbf{A}_{\boldsymbol{i} \circ \boldsymbol{r}} \mathbf{B}_{\boldsymbol{j} \circ \boldsymbol{l}} \mathbf{B}_{\boldsymbol{j} \circ \boldsymbol{r}}
$$

$$
= \frac{1}{b} \sum_{\boldsymbol{i}, \boldsymbol{j} \in [n]^q} \langle \mathbf{A}(\boldsymbol{e}_{i_1}, \cdots, \boldsymbol{e}_{i_q}, \mathbf{I}, \cdots, \mathbf{I}), \mathbf{B}(\boldsymbol{e}_{j_1}, \cdots, \boldsymbol{e}_{j_q}, \mathbf{I}, \cdots, \mathbf{I}) \rangle^2
$$

$$
\leq \frac{1}{b} \sum_{\boldsymbol{i}, \boldsymbol{j} \in [n]^q} \|\mathbf{A}(\boldsymbol{e}_{i_1}, \cdots, \boldsymbol{e}_{i_q}, \mathbf{I}, \cdots, \mathbf{I})\|_F^2 \|\mathbf{B}(\boldsymbol{e}_{j_1}, \cdots, \boldsymbol{e}_{j_q}, \mathbf{I}, \cdots, \mathbf{I})\|_F^2
$$

$$
= \frac{\|\mathbf{A}\|_F^2 \|\mathbf{B}\|_F^2}{b}. \tag{40}
$$

Here $\circ$ denotes concatenation, that is, $\boldsymbol{i} \circ \boldsymbol{l} = (i_1, \cdots, i_q, l_1, \cdots, l_{p-q}) \in [n]^p$. The fourth equation is Cauchy-Schwartz inequality. Finally note that there are no more than $4^p$ ways of assigning $q$ positions to $\boldsymbol{l}$ and $\boldsymbol{l}'$ each. Combining Eq. (39) and (40) we get

$$
\mathbb{V}[\langle \tilde{\boldsymbol{s}}_{\mathbf{A}}, \tilde{\boldsymbol{s}}_{\widetilde{\mathbf{B}}} \rangle] = \mathbb{E}[\langle \tilde{\boldsymbol{s}}_{\mathbf{A}}, \tilde{\boldsymbol{s}}_{\widetilde{\mathbf{B}}} \rangle^2] - \langle \mathbf{A}, \mathbf{B} \rangle^2 \leq \frac{4^p \|\mathbf{A}\|_F^2 \|\mathbf{B}\|_F^2}{b},
$$

which completes the proof. $\qquad\square$

*Proof of Lemma 2.* Eq. (32) immediately follows Eq. (28) by adding everything together. For the variance bound we cannot use the same argument because in general the $m^2$ random variables are neither independent nor uncorrelated. Instead, we compute the variance by definition. First we compute the expected square term as follows:

$$
\mathbb{E}\left[\left(\sum_{i,j} w_i w_j \langle \tilde{\boldsymbol{s}}_{\mathbf{A}_i}, \tilde{\boldsymbol{s}}_{\widetilde{\mathbf{B}}_j} \rangle\right)^2\right]
$$

$$
= \sum_{\substack{i,j,i',j' \\ \boldsymbol{l}, \boldsymbol{l}', \boldsymbol{r}, \boldsymbol{r}'}} w_i w_j w_{i'} w_{j'} \cdot \mathbb{E}[\delta(\boldsymbol{l}, \boldsymbol{l}') \delta(\boldsymbol{r}, \boldsymbol{r}')] \cdot \mathbb{E}[\sigma(\boldsymbol{l}) \bar{\sigma}(\boldsymbol{l}') \bar{\sigma}(\boldsymbol{r}) \sigma(\boldsymbol{r}')] \cdot [\mathbf{A}_i]_{\boldsymbol{l}} [\mathbf{A}_{i'}]_{\boldsymbol{r}} [\widetilde{\mathbf{B}}_j]_{\boldsymbol{l}'} [\widetilde{\mathbf{B}}_{j'}]_{\boldsymbol{r}'}. \tag{41}
$$

Define $\mathbf{X} = \sum_i w_i \mathbf{A}_i$ and $\mathbf{Y} = \sum_i w_i \mathbf{B}_i$. The above equation can then be simplified as

$$
\mathbb{E}\left[\left(\sum_{i,j} w_i w_j \langle \tilde{\boldsymbol{s}}_{\mathbf{A}_i}, \tilde{\boldsymbol{s}}_{\widetilde{\mathbf{B}}_j} \rangle\right)^2\right] = \sum_{\boldsymbol{l}, \boldsymbol{l}', \boldsymbol{r}, \boldsymbol{r}'} \mathbb{E}[\delta(\boldsymbol{l}, \boldsymbol{l}') \delta(\boldsymbol{r}, \boldsymbol{r}')] \cdot \mathbb{E}[\sigma(\boldsymbol{l}) \bar{\sigma}(\boldsymbol{l}') \bar{\sigma}(\boldsymbol{r}) \sigma(\boldsymbol{r}')] \cdot \mathbf{X}_{\boldsymbol{l}} \mathbf{X}_{\boldsymbol{r}} \widetilde{\mathbf{Y}}_{\boldsymbol{l}'} \widetilde{\mathbf{Y}}_{\boldsymbol{r}'}. \tag{42}
$$

Applying Lemma 1 we have

$$
\mathbb{V}\left[\sum_{i,j} w_i w_j \langle \tilde{\boldsymbol{s}}_{\mathbf{A}_i}, \tilde{\boldsymbol{s}}_{\widetilde{\mathbf{B}}_j} \rangle\right] \leq \frac{4^p \|\mathbf{X}\|_F^2 \|\mathbf{Y}\|_F^2}{b}. \tag{43}
$$

Finally, note that

$$
\|\mathbf{X}\|_F^2 = \sum_{i,j} w_i w_j \langle \mathbf{A}_i, \mathbf{A}_j \rangle \leq \sum_{i,j} w_i w_j \|\mathbf{A}_i\|_F \|\mathbf{A}_j\|_F \leq \|\boldsymbol{w}\|^2 \max_i \|\mathbf{A}_i\|_F^2. \tag{44}
$$

$\square$

With Lemma 1 and 2, we can easily prove Theorem 1.

*Proof of Theorem 1.* First we prove the $\varepsilon_1(\boldsymbol{u})$ bound. Let $\mathbf{A} = \mathbf{T}$ and $\mathbf{B} = \boldsymbol{u}^{\otimes 3}$. Note that $\|\mathbf{A}\|_F = \|\mathbf{T}\|_F$ and $\|\mathbf{B}\|_F = \|\boldsymbol{u}\|^2 = 1$. Note that $[\mathbf{T}(\mathbf{I}, \boldsymbol{u}, \boldsymbol{u})]_i = \mathbf{T}(\boldsymbol{e}_i, \boldsymbol{u}, \boldsymbol{u})$. Next we consider $\varepsilon_2(\boldsymbol{u})$ and let $\mathbf{A} = \mathbf{T}$, $\mathbf{B} = \boldsymbol{e}_i \otimes \boldsymbol{u} \otimes \boldsymbol{u}$. Again we have $\|\mathbf{A}\|_F = \|\mathbf{T}\|_F$ and $\|\mathbf{B}\|_F = 1$. A union bound over all $i = 1, \cdots, n$ yields the result. For the inequality involving $\boldsymbol{w}$ we apply Lemma 2. $\square$

### E.3 Analysis of fast robust tensor power method

In this section, we prove Theorem 3, a more refined version of Theorem 2 in Section 4.2. We structure the section by first demonstrating the convergence behavior of noisy tensor power method, and then show how error accumulates with deflation. Finally, the overall bound is derived by combining these two parts.

#### E.3.1 Recovering the principal eigenvector

Define the angle between two vectors $\boldsymbol{v}$ and $\boldsymbol{u}$ to be $\theta(\boldsymbol{v}, \boldsymbol{u})$. First, in Lemma 3 we show that if the initialization vector $\boldsymbol{u}_0$ is randomly chosen from the unit sphere, then the angle $\theta$ between the iteratively updated vector $\boldsymbol{u}_t$ and the largest eigenvector of tensor $\mathbf{T}$, $\boldsymbol{v}_1$, will decrease to a point that $\tan \theta(\boldsymbol{v}_1, \boldsymbol{u}_t) < 1$. Afterwards, in Lemma 4, we use a similar approach as in [35] to prove that the error between the final estimation and the ground truth is bounded.

Suppose $\mathbf{T}$ is the exact low-rank ground truth tensor and Each noisy tensor update can then be written as

$$\tilde{\boldsymbol{u}}_{t+1} = \mathbf{T}(\mathbf{I}, \boldsymbol{u}_t, \boldsymbol{u}_t) + \tilde{\varepsilon}(\boldsymbol{u}_t), \tag{45}$$

where $\tilde{\varepsilon}(\boldsymbol{u}_t) = \mathbf{E}(\mathbf{I}, \boldsymbol{u}_t, \boldsymbol{u}_t) + \varepsilon_{2,T}(\boldsymbol{u}_t)$ is the noise coming from statistical and tensor sketch approximation error.

Before presenting key lemmas, we first define $\gamma$-*separation*, a concept introduced in [1].

**Definition 1** ($\gamma$-separation, [1]). *Fix $i^* \in [k]$, $\boldsymbol{u} \in \mathbb{R}^n$ and $\gamma > 0$. $u$ is $\gamma$-separated with respect to $\boldsymbol{v}_{i^*}$ if the following holds:*

$$\lambda_{i^*}\langle \boldsymbol{u}, \boldsymbol{v}_{i^*}\rangle - \max_{i \in [k]\setminus\{i^*\}} \lambda_i\langle \boldsymbol{u}, \boldsymbol{v}_i\rangle \geq \gamma\lambda_{i^*}\langle \boldsymbol{u}, \boldsymbol{v}_{i^*}\rangle. \tag{46}$$

Lemma 3 analyzes the first phase of the noisy tensor power algorithm. It shows that if the initialization vector $\boldsymbol{u}_0$ is $\gamma$-separated with respect to $\boldsymbol{v}_1$ and the magnitude of noise $\tilde{\varepsilon}(\boldsymbol{u}_t)$ is small at each iteration $t$, then after a short number of iterations we will have inner product between $\boldsymbol{u}_t$ and $\boldsymbol{v}_1$ at least a constant.

**Lemma 3.** *Let $\{\boldsymbol{v}_1, \boldsymbol{v}_2, \cdots, \boldsymbol{v}_k\}$ and $\{\lambda_1, \lambda_2, \cdots, \lambda_k\}$ be eigenvectors and eigenvalues of tensor $\mathbf{T} \in \mathbb{R}^{n \times n \times n}$, where $\lambda_1 |\langle \boldsymbol{v}_1, \boldsymbol{u}_0\rangle| = \max_{i \in [k]} \lambda_i |\langle \boldsymbol{v}_i, \boldsymbol{u}_0\rangle|$. Denote $\mathbf{V} = (\boldsymbol{v}_1, \cdots, \boldsymbol{v}_k) \in \mathbb{R}^{n \times k}$ as the matrix for eigenvectors. Suppose that for every iteration $t$ the noise satisfies*

$$|\langle \boldsymbol{v}_i, \tilde{\varepsilon}(\boldsymbol{u}_t)\rangle| \leq \epsilon_1 \;\; \forall i \in [n] \;\; and \;\; \|\mathbf{V}^\top \tilde{\varepsilon}(\boldsymbol{u}_t)\| \leq \epsilon_2; \tag{47}$$

*suppose also the initialization $\boldsymbol{u}_0$ is $\gamma$-separated with respect to $\boldsymbol{v}_1$ for some $\gamma \in (0.5, 1)$. If $\tan \theta(\boldsymbol{v}_1, \boldsymbol{u}_0) > 1$, and*

$$\epsilon_1 \leq \min\left( \frac{1}{4\frac{\max_{i \in [k]} \lambda_i}{\lambda_1} + 2}, \frac{1 - (1 + \alpha)/2}{2} \right) \lambda_1 \langle \boldsymbol{v}_1, \boldsymbol{u}_0\rangle^2 \;\; and \;\; \epsilon_2 \leq \frac{1 - (1 + \alpha)/2}{2\sqrt{2}(1 + \alpha)} \lambda_1 |\langle \boldsymbol{v}_1, \boldsymbol{u}_0\rangle|$$

$$\tag{48}$$

*for some $\alpha > 0$, then for a small constant $\rho > 0$, there exists a $T > \log_{1+\alpha}(1 + \rho)\tan \theta(\boldsymbol{v}_1, \boldsymbol{u}_0)$ such that after $T$ iteration, we have $\tan \theta(\boldsymbol{v}_1, \boldsymbol{u}_T) < \frac{1}{1+\rho}$,*

*Proof.* Let $\tilde{\boldsymbol{u}}_{t+1} = \mathbf{T}\left(\mathbf{I}, \boldsymbol{u}_t, \boldsymbol{u}_t\right) + \tilde{\boldsymbol{\varepsilon}}(\boldsymbol{u}_t)$ and $\boldsymbol{u}_{t+1} = \tilde{\boldsymbol{u}}_{t+1} / \|\tilde{\boldsymbol{u}}_{t+1}\|$. For $\alpha \in (0,1)$, we try to prove that there exists a $T$ such that for $t > T$

$$\frac{1}{\tan\theta\left(\boldsymbol{v}_1, \boldsymbol{u}_{t+1}\right)} = \frac{|\langle \boldsymbol{v}_1, \boldsymbol{u}_{t+1}\rangle|}{\left(1 - \langle \boldsymbol{v}_1, \boldsymbol{u}_{t+1}\rangle^2\right)^{1/2}} = \frac{|\langle \boldsymbol{v}_1, \tilde{\boldsymbol{u}}_{t+1}\rangle|}{\left(\sum\limits_{i=2}^{n} \langle \boldsymbol{v}_i, \tilde{\boldsymbol{u}}_{t+1}\rangle^2\right)^{1/2}} \geq 1. \tag{49}$$

First we examine the numerator. Using the assumption $|\langle \boldsymbol{v}_i, \tilde{\boldsymbol{\varepsilon}}(\boldsymbol{u}_t)\rangle| \leq \epsilon_1$ and the fact that $\langle \boldsymbol{v}_i, \tilde{\boldsymbol{u}}_{t+1}\rangle = \lambda_i \langle \boldsymbol{v}_i, \boldsymbol{u}_t\rangle^2 + \langle \boldsymbol{v}_i, \tilde{\boldsymbol{\varepsilon}}(\boldsymbol{u}_t)\rangle$, we have

$$|\langle \boldsymbol{v}_i, \tilde{\boldsymbol{u}}_{t+1}\rangle| \geq \lambda_i \langle \boldsymbol{v}_i, \boldsymbol{u}_t\rangle^2 - \epsilon_1 \geq |\langle \boldsymbol{v}_i, \boldsymbol{u}_t\rangle| \left(\lambda_i |\langle \boldsymbol{v}_i, \boldsymbol{u}_t\rangle| - \epsilon_1 / |\langle \boldsymbol{v}_i, \boldsymbol{u}_t\rangle|\right). \tag{50}$$

For the denominator, by Hölder's inequality we have

$$\left(\sum_{i=2}^{n} \langle \boldsymbol{v}_i, \tilde{\boldsymbol{u}}_{t+1}\rangle^2\right)^{1/2} = \left(\sum_{i=2}^{n} \left(\lambda_i \langle \boldsymbol{v}_i, \boldsymbol{u}_t\rangle^2 + \langle \boldsymbol{v}_i, \tilde{\boldsymbol{\varepsilon}}(\boldsymbol{u}_t)\rangle\right)^{1/2}\right) \tag{51}$$

$$\leq \left(\sum_{i=2}^{n} \lambda_i^2 \langle \boldsymbol{v}_i, \boldsymbol{u}_t\rangle^4\right)^{1/2} + \left(\sum_{i=2}^{n} \langle \boldsymbol{v}_i, \tilde{\boldsymbol{\varepsilon}}(\boldsymbol{u}_t)\rangle^2\right)^{1/2} \tag{52}$$

$$\leq \max_{i \neq 1} \lambda_i |\langle \boldsymbol{v}_i, \boldsymbol{u}_t\rangle| \left(\sum_{i=2}^{n} \langle \boldsymbol{v}_i, \boldsymbol{u}_t\rangle^2\right)^{1/2} + \epsilon_2 \tag{53}$$

$$\leq \left(1 - \langle \boldsymbol{v}_1, \boldsymbol{u}_t\rangle^2\right)^{1/2} \left(\max_{i \neq 1} \lambda_i |\langle \boldsymbol{v}_i, \boldsymbol{u}_t\rangle| + \epsilon_2 / \left(1 - \langle \boldsymbol{v}_1, \boldsymbol{u}_t\rangle^2\right)^{1/2}\right) \tag{54}$$

Equation (50) and (51) yield

$$\frac{1}{\tan\theta\left(\boldsymbol{v}_1, \boldsymbol{u}_{t+1}\right)} \geq \frac{|\langle \boldsymbol{v}_1, \boldsymbol{u}_t\rangle|}{\left(1 - \langle \boldsymbol{v}_1, \boldsymbol{u}_t\rangle^2\right)^{1/2}} \frac{\lambda_1 |\langle \boldsymbol{v}_1, \boldsymbol{u}_t\rangle| - \epsilon_1 / |\langle \boldsymbol{v}_1, \boldsymbol{u}_t\rangle|}{\max\limits_{i \neq 1} \lambda_i |\langle \boldsymbol{v}_i, \boldsymbol{u}_t\rangle| + \epsilon_2 / \left(1 - \langle \boldsymbol{v}_1, \boldsymbol{u}_t\rangle^2\right)^{1/2}} \tag{55}$$

$$= \frac{1}{\tan\theta\left(\boldsymbol{v}_1, \boldsymbol{u}_t\right)} \frac{\lambda_1 |\langle \boldsymbol{v}_1, \boldsymbol{u}_t\rangle| - \epsilon_1 / |\langle \boldsymbol{v}_1, \boldsymbol{u}_t\rangle|}{\max\limits_{i \neq 1} \lambda_i |\langle \boldsymbol{v}_i, \boldsymbol{u}_t\rangle| + \epsilon_2 / \left(1 - \langle \boldsymbol{v}_1, \boldsymbol{u}_t\rangle^2\right)^{1/2}} \tag{56}$$

To prove that the second term is larger than $1 + \alpha$, we first show that when $t = 0$, the inequality holds. Since the initialization vector is a $\gamma-$separated vector, we have

$$\lambda_1 |\langle \boldsymbol{v}_1, \boldsymbol{u}_0\rangle| - \max_{i \in [k]} \lambda_i |\langle \boldsymbol{v}_i, \boldsymbol{u}_0\rangle| \geq \gamma \lambda_1 |\langle \boldsymbol{v}_1, \boldsymbol{u}_0\rangle|, \tag{57}$$

$$\max_{i \in [k]} \lambda_i |\langle \boldsymbol{v}_i, \boldsymbol{u}_0\rangle| \leq (1 - \gamma)\lambda_1 |\langle \boldsymbol{v}_1, \boldsymbol{u}_0\rangle| \leq 0.5\lambda_1 |\langle \boldsymbol{v}_1, \boldsymbol{u}_0\rangle|, \tag{58}$$

the last inequality holds since $\gamma > 0.5$. Note that we assume $\tan\theta\left(\boldsymbol{v}_1, \boldsymbol{u_0}\right) > 1$ and hence $\langle \boldsymbol{v}_1, \boldsymbol{u}_0\rangle^2 < 0.5$. Therefore,

$$\epsilon_2 \leq \frac{1 - (1 + \alpha)/2}{2\sqrt{2}(1 + \alpha)} \lambda_1 |\langle \boldsymbol{v}_1, \boldsymbol{u}_0\rangle| \leq \frac{\left(1 - \langle \boldsymbol{v}_1, \boldsymbol{u}_0\rangle^2\right)^{1/2} (1 - (1 + \alpha)/2)}{2(1 + \alpha)} \lambda_1 |\langle \boldsymbol{v}_1, \boldsymbol{u}_0\rangle|. \tag{59}$$

Thus, for $t = 0$, using the condition for $\epsilon_1$ and $\epsilon_2$ we have

$$\frac{\lambda_1 |\langle \boldsymbol{v}_i, \boldsymbol{u}_0\rangle| - \epsilon_1 / |\langle \boldsymbol{v}_i, \boldsymbol{u}_0\rangle|}{\max\limits_{i \neq 1} \lambda_i |\langle \boldsymbol{v}_i, \boldsymbol{u}_0\rangle| + \epsilon_2 / \left(1 - \langle \boldsymbol{v}_1, \boldsymbol{u}_0\rangle^2\right)^{1/2}} \geq \frac{\lambda_1 |\langle \boldsymbol{v}_i, \boldsymbol{u}_0\rangle| - \epsilon_1 / |\langle \boldsymbol{v}_i, \boldsymbol{u}_0\rangle|}{0.5\lambda_1 |\langle \boldsymbol{v}_1, \boldsymbol{u}_0\rangle| + \epsilon_2 / \left(1 - \langle \boldsymbol{v}_1, \boldsymbol{u}_0\rangle^2\right)^{1/2}} \geq 1 + \alpha. \tag{60}$$

The result yields $1 / \tan\theta\left(\boldsymbol{v}_1, \boldsymbol{u}_1\right) > (1 + \alpha) / \tan\theta\left(\boldsymbol{v}_1, \boldsymbol{u}_0\right)$. This also indicates that $|\langle \boldsymbol{v}_1, \boldsymbol{u}_1\rangle| > |\langle \boldsymbol{v}_1, \boldsymbol{u}_0\rangle|$, which implies that

$$\epsilon_1 \leq \min\left(\frac{1}{4\frac{\max_{i \in [k]} \lambda_i}{\lambda_1} + 2}, \frac{1 - (1 + \alpha)/2}{2}\right) \lambda_1 \langle \boldsymbol{v}_1, \boldsymbol{u}_t\rangle^2 \text{ and } \epsilon_2 \leq \frac{1 - (1 + \alpha)/2}{2\sqrt{2}(1 + \alpha)} \lambda_1 |\langle \boldsymbol{v}_1, \boldsymbol{u}_t\rangle| \tag{61}$$

also holds for $t = 1$. Next we need to make sure that for $t \geq 0$

$$\max_{i \neq 1} \lambda_i |\langle \boldsymbol{v}_i, \boldsymbol{u}_t \rangle| \leq 0.5 \lambda_1 |\langle \boldsymbol{v}_1, \boldsymbol{u}_t \rangle|. \tag{62}$$

In other words, we need to show that $\frac{\lambda_1 |\langle \boldsymbol{v}_1, \boldsymbol{u}_t \rangle|}{\max_{i \neq 1} \lambda_i |\langle \boldsymbol{v}_i, \boldsymbol{u}_t \rangle|} \geq 2$. From Equation (58), for $t = 0$,

$\frac{\lambda_1 |\langle \boldsymbol{v}_1, \boldsymbol{u}_t \rangle|}{\max_{i \neq 1} \lambda_i |\langle \boldsymbol{v}_i, \boldsymbol{u}_t \rangle|} \geq \frac{1}{1-\gamma} \geq 2$. For every $i \in [k]$,

$$|\langle \boldsymbol{v}_i, \tilde{\boldsymbol{u}}_{t+1} \rangle| \leq \lambda_i |\langle \boldsymbol{v}_i, \boldsymbol{u}_t \rangle|^2 + \epsilon_1 \leq |\langle \boldsymbol{v}_i, \boldsymbol{u}_t \rangle| \left( \lambda_i |\langle \boldsymbol{v}_i, \boldsymbol{u}_t \rangle| + \epsilon_1 / |\langle \boldsymbol{v}_i, \boldsymbol{u}_t \rangle| \right). \tag{63}$$

With equation (50), we have

$$\frac{\lambda_1 |\langle \boldsymbol{v}_1, \boldsymbol{u}_{t+1} \rangle|}{\lambda_i |\langle \boldsymbol{v}_i, \boldsymbol{u}_{t+1} \rangle|} = \frac{\lambda_1 |\langle \boldsymbol{v}_1, \tilde{\boldsymbol{u}}_{t+1} \rangle|}{\lambda_i |\langle \boldsymbol{v}_i, \tilde{\boldsymbol{u}}_{t+1} \rangle|} \geq \frac{\lambda_1 |\langle \boldsymbol{v}_1, \boldsymbol{u}_t \rangle| \left( \lambda_1 |\langle \boldsymbol{v}_1, \boldsymbol{u}_t \rangle| - \frac{\epsilon_1}{|\langle \boldsymbol{v}_1, \boldsymbol{u}_t \rangle|} \right)}{\lambda_i |\langle \boldsymbol{v}_i, \boldsymbol{u}_t \rangle| \left( \lambda_i |\langle \boldsymbol{v}_i, \boldsymbol{u}_t \rangle| - \frac{\epsilon_1}{|\langle \boldsymbol{v}_i, \boldsymbol{u}_t \rangle|} \right)} \tag{64}$$

$$= \left( \frac{\lambda_1 |\langle \boldsymbol{v}_1, \boldsymbol{u}_t \rangle|}{\lambda_i |\langle \boldsymbol{v}_i, \boldsymbol{u}_t \rangle|} \right)^2 \frac{1 - \frac{\epsilon_1}{\lambda_1 \langle \boldsymbol{v}_1, \boldsymbol{u}_t \rangle^2}}{1 + \frac{\lambda_i}{\lambda_1} \frac{\epsilon_1}{\lambda_1 \langle \boldsymbol{v}_1, \boldsymbol{u}_t \rangle^2} \left( \frac{\lambda_1 |\langle \boldsymbol{v}_1, \boldsymbol{u}_t \rangle|}{\lambda_i |\langle \boldsymbol{v}_i, \boldsymbol{u}_t \rangle|} \right)^2} \tag{65}$$

$$\geq \left( \frac{\lambda_1 |\langle \boldsymbol{v}_1, \boldsymbol{u}_t \rangle|}{\lambda_i |\langle \boldsymbol{v}_i, \boldsymbol{u}_t \rangle|} \right)^2 \frac{1 - \frac{\epsilon_1}{\lambda_1 \langle \boldsymbol{v}_1, \boldsymbol{u}_t \rangle^2}}{1 + \frac{\max_{i \in [k]} \lambda_i}{\lambda_1} \frac{\epsilon_1}{\lambda_1 \langle \boldsymbol{v}_1, \boldsymbol{u}_t \rangle^2} \left( \frac{\lambda_1 |\langle \boldsymbol{v}_1, \boldsymbol{u}_t \rangle|}{\lambda_i |\langle \boldsymbol{v}_i, \boldsymbol{u}_t \rangle|} \right)^2} \tag{66}$$

$$= \frac{1 - \frac{\epsilon_1}{\lambda_1 \langle \boldsymbol{v}_1, \boldsymbol{u}_t \rangle^2}}{\left( \frac{\lambda_1 |\langle \boldsymbol{v}_1, \boldsymbol{u}_t \rangle|}{\lambda_i |\langle \boldsymbol{v}_i, \boldsymbol{u}_t \rangle|} \right)^{-2} + \frac{\max_{i \in [k]} \lambda_i}{\lambda_1} \frac{\epsilon_1}{\lambda_1 \langle \boldsymbol{v}_1, \boldsymbol{u}_t \rangle^2}}. \tag{67}$$

Let $\kappa = \frac{\max_{i \in [k]} \lambda_i}{\lambda_1}$. For $t = 0$, with conditions on $\epsilon_1$ the following holds:

$$\frac{\lambda_1 |\langle \boldsymbol{v}_1, \boldsymbol{u}_1 \rangle|}{\lambda_i |\langle \boldsymbol{v}_i, \boldsymbol{u}_1 \rangle|} \geq \frac{1 - \frac{\epsilon_1}{\lambda_1 \langle \boldsymbol{v}_1, \boldsymbol{u}_0 \rangle^2}}{\left( \frac{\lambda_1 |\langle \boldsymbol{v}_1, \boldsymbol{u}_0 \rangle|}{\lambda_i |\langle \boldsymbol{v}_i, \boldsymbol{u}_0 \rangle|} \right)^{-2} + \frac{\max_{i \in [k]} \lambda_i}{\lambda_1} \frac{\epsilon_1}{\lambda_1 \langle \boldsymbol{v}_1, \boldsymbol{u}_0 \rangle^2}}. \tag{68}$$

$$\geq \frac{1 - \frac{1}{4\kappa + 2}}{\frac{1}{4} + \frac{\kappa}{4\kappa + 2}} = 2 \tag{69}$$

With the two conditions stated in Equation (61), following the same step in (60), we have $\frac{1}{\tan \theta(\boldsymbol{v}_1, u_2)} \geq (1 + \alpha) \frac{1}{\tan \theta(\boldsymbol{v}_1, u_1)}$. By induction, $\frac{1}{\tan \theta(\boldsymbol{v}_1, u_{t+1})} \geq (1 + \alpha) \frac{1}{\tan \theta(\boldsymbol{v}_1, t)}$. for $t \geq 0$. Subsequently,

$$\frac{1}{\tan \theta(\boldsymbol{v}_1, u_T)} \geq (1 + \alpha)^T \frac{1}{\tan \theta(\boldsymbol{v}_1, \boldsymbol{u}_0)}. \tag{70}$$

Finally, we complete the proof by setting $T > \log_{1+\alpha}(1 + \rho) \tan \theta(\boldsymbol{v}_1, \boldsymbol{u}_0)$. $\qquad \square$

Next, we present Lemma 4, which analyzes the second phase of the noisy tensor power method. The second phase starts with $\tan \theta(\boldsymbol{v}_1, \boldsymbol{u}_0) < 1$, that is, the inner product of $\boldsymbol{v}_1$ and $\boldsymbol{u}_0$ is lower bounded by $1/2$.

**Lemma 4.** *Let $\boldsymbol{v}_1$ be the principal eigenvector of a tensor $\mathbf{T}$ and let $\boldsymbol{u}_0$ be an arbitrary vector in $\mathbb{R}^d$ that satisfies $\tan \theta(\boldsymbol{v}_1, \boldsymbol{u}_0) < 1$. Suppose at every iteration $t$ the noise satisfies*

$$4 \|\tilde{\varepsilon}(\boldsymbol{u}_t)\| \leq \epsilon(\lambda_1 - \lambda_2) \quad \text{and} \quad 4 |\langle \boldsymbol{v}_1, \tilde{\varepsilon}(\boldsymbol{u}_t) \rangle| \leq (\lambda_1 - \lambda_2) \cos^2 \theta(\boldsymbol{v}_1, \boldsymbol{u}_0) \tag{71}$$

*for some $\epsilon < 1$. Then with high probability there exists $T = O\left( \frac{\lambda_1}{\lambda_1 - \lambda_2} \log(1/\epsilon) \right)$ such that after $T$ iteration we have $\tan \theta(\boldsymbol{v}_1, \boldsymbol{u}_T) \leq \epsilon$.*

*Proof.* Define $\Delta := \frac{\lambda_1 - \lambda_2}{4}$ and $\mathbf{X} := \boldsymbol{v}_1^\perp$. We have the following chain of inequalities:

$$\tan \theta(\boldsymbol{v}_1, \mathbf{T}(\mathbf{I}, \boldsymbol{u}, \boldsymbol{u}) + \tilde{\varepsilon}(\boldsymbol{u})) \leq \frac{\|\mathbf{X}^T(\mathbf{T}(\mathbf{I}, \boldsymbol{u}, \boldsymbol{u}) + \tilde{\varepsilon}(\boldsymbol{u}))\|}{\|\boldsymbol{v}_1^T(\mathbf{T}(\mathbf{I}, \boldsymbol{u}, \boldsymbol{u}) + \tilde{\varepsilon}(\boldsymbol{u}))\|} \tag{72}$$

$$\leq \frac{\left\|\mathbf{X}^T \mathbf{T}\left(\mathbf{I}, \boldsymbol{u}, \boldsymbol{u}\right)\right\| + \left\|\mathbf{V}^T \tilde{\varepsilon}(\boldsymbol{u})\right\|}{\left\|\boldsymbol{v}_1^T \mathbf{T}\left(\mathbf{I}, \boldsymbol{u}, \boldsymbol{u}\right)\right\| - \left\|\boldsymbol{v}_1^T \tilde{\varepsilon}(\boldsymbol{u})\right\|} \tag{73}$$

$$\leq \frac{\lambda_2 \left\|\mathbf{X}^T \boldsymbol{u}\right\|^2 + \|\tilde{\varepsilon}(\boldsymbol{u})\|}{\lambda_1 \left|\boldsymbol{v}_1^T \boldsymbol{u}\right|^2 - \left|\boldsymbol{v}_1^\top \tilde{\varepsilon}(\boldsymbol{u})\right|} \tag{74}$$

$$= \frac{\left\|\mathbf{X}^T \boldsymbol{u}\right\|^2}{\left|\boldsymbol{v}_1^T \boldsymbol{u}\right|^2} \frac{\lambda_2}{\lambda_1 - \frac{\left|\boldsymbol{v}_1^\top \tilde{\varepsilon}(\boldsymbol{u})\right|}{\left|\boldsymbol{v}_1^\top \boldsymbol{u}\right|^2}} + \frac{\frac{\|\tilde{\varepsilon}(\boldsymbol{u})\|}{\left|\boldsymbol{v}_1^\top \boldsymbol{u}\right|^2}}{\lambda_1 - \frac{\left|\boldsymbol{v}_1^\top \tilde{\varepsilon}(\boldsymbol{u})\right|}{\left|\boldsymbol{v}_1^\top \boldsymbol{u}\right|^2}} \tag{75}$$

$$\leq \tan^2 \theta(\boldsymbol{v}_1, \boldsymbol{u}) \frac{\lambda_2}{\lambda_2 + 3\Delta} + \frac{\Delta\epsilon \left(1 + \tan^2 \theta\left(\boldsymbol{v}_1, \boldsymbol{u}\right)\right)}{\lambda_2 + 3\Delta} \tag{76}$$

$$\leq \max\left(\epsilon, \frac{\lambda_2 + \Delta\epsilon}{\lambda_2 + 2\Delta} \tan^2 \theta\left(\boldsymbol{v}_1, \boldsymbol{u}\right)\right) \tag{77}$$

$$\leq \max\left(\epsilon, \frac{\lambda_2 + \Delta\epsilon}{\lambda_2 + 2\Delta} \tan \theta\left(\boldsymbol{v}_1, \boldsymbol{u}\right)\right) \tag{78}$$

The second step follows by triangle inequality. For $\boldsymbol{u} = \boldsymbol{u}_0$, using the condition $\tan\left(\boldsymbol{v}_1, \boldsymbol{u}_0\right) < 1$ we obtain

$$\tan \theta\left(\boldsymbol{v}_1, \boldsymbol{u}_1\right) \leq \max\left(\epsilon, \frac{\lambda_2 + \Delta\epsilon}{\lambda_2 + 2\Delta} \tan^2 \theta\left(\boldsymbol{v}_1, \boldsymbol{u}\right)\right) \leq \max\left(\epsilon, \frac{\lambda_2 + \Delta\epsilon}{\lambda_2 + 2\Delta} \tan \theta\left(\boldsymbol{v}_1, \boldsymbol{u}\right)\right) \tag{79}$$

Since $\frac{\lambda_2 + \Delta\epsilon}{\lambda_2 + 2\Delta} \leq \max\left(\frac{\lambda_2}{\lambda_2 + \Delta}, \epsilon\right) \leq (\lambda_2/\lambda_1)^{1/4} < 1$, we have

$$\tan \theta\left(\boldsymbol{v}_1, \boldsymbol{u}_1\right) = \tan \theta\left(\boldsymbol{v}_1, \mathbf{T}\left(\mathbf{I}, \boldsymbol{u}_0, \boldsymbol{u}_0\right) + \tilde{\varepsilon}(\boldsymbol{u}_t)\right) \leq \max\left(\epsilon, (\lambda_2/\lambda_1)^{1/4} \tan \theta\left(\boldsymbol{v}_1, \boldsymbol{u}_0\right)\right) < 1. \tag{80}$$

By induction,

$$\tan \theta\left(\boldsymbol{v}_1, \boldsymbol{u}_{t+1}\right) = \tan \theta\left(\boldsymbol{v}_1, \mathbf{T}\left(\mathbf{I}, \boldsymbol{u}_t, \boldsymbol{u}_t\right) + \tilde{\varepsilon}(\boldsymbol{u}_t)\right) \leq \max\left(\epsilon, (\lambda_2/\lambda_1)^{1/4} \tan \theta\left(\boldsymbol{v}_1, \boldsymbol{u}_t\right)\right) < 1.$$

for every $t$. Eq. (78) then yields

$$\tan \theta\left(\boldsymbol{v}_1, \boldsymbol{u}_T\right) \leq \max\left(\epsilon, \max \epsilon, (\lambda_2/\lambda_1)^{L/4} \tan \theta\left(\boldsymbol{v}_1, \boldsymbol{u}_0\right)\right). \tag{81}$$

Consequently, after $T = \log_{(\lambda_2/\lambda_1)^{-1/4}}(1/\epsilon)$ iterations we have $\tan \theta\left(\boldsymbol{v}_1, \boldsymbol{u}_T\right) \leq \epsilon$. $\qquad \square$

**Lemma 5.** *Suppose $\boldsymbol{v}_1$ is the principal eigenvector of a tensor $\mathbf{T}$ and let $\boldsymbol{u}_0 \in \mathbb{R}^n$. For some $\alpha, \rho > 0$ and $\epsilon < 1$, if at every step, the noise satisfies*

$$\|\tilde{\varepsilon}(\boldsymbol{u}_t)\| \leq \epsilon \frac{\lambda_1 - \lambda_2}{4} \quad \text{and} \quad \left|\langle \boldsymbol{v}_1, \tilde{\varepsilon}(\boldsymbol{u}_t)\rangle\right| \leq \min\left(\frac{1}{4\frac{\max_{i \in [k]} \lambda_i}{\lambda_1} + 2}\lambda_1, \frac{1 - (1+\alpha)/2}{2\sqrt{2}(1+\alpha)}\lambda_1\right) \frac{1}{\tau^2 n}, \tag{82}$$

*then with high probability there exists an $T = O\left(\log_{1+\alpha}(1+\rho)\tau\sqrt{n} + \frac{\lambda_1}{\lambda_1 - \lambda_2}\log(1/\epsilon)\right)$ such that after $T$ iterations we have $\left\|\left(\boldsymbol{I} - \boldsymbol{u}_T \boldsymbol{u}_T^T\right)\boldsymbol{v}_1\right\| \leq \epsilon$.*

*Proof.* By Lemma 2.5 in [35], for any fixed orthonormal matrix $\mathbf{V}$ and a random vector $\boldsymbol{u}$, we have $\max_{i \in [K]} \tan \theta(\boldsymbol{v}_i, \boldsymbol{u}) \leq \tau\sqrt{n}$ with all but $O(\tau^{-1} + e^{-\Omega(d)})$ probability. Using the fact that $\cos \theta\left(\boldsymbol{v}_1, \boldsymbol{u}_0\right) \geq 1/(1 + \tan \theta\left(\boldsymbol{v}_1, \boldsymbol{u}_0\right)) \geq \frac{1}{\tau\sqrt{n}}$, the following bounds on the noise level imply the conditions in Lemma 3:

$$\left\|\mathbf{V}^T \tilde{\varepsilon}(\boldsymbol{u}_t)\right\| \leq \frac{1 - (1+\alpha)/2}{2\sqrt{2}(1+\alpha)\tau\sqrt{n}} \quad \text{and} \quad \left|\langle \boldsymbol{v}_1, \tilde{\varepsilon}(\boldsymbol{u}_t)\rangle\right|$$

$$\leq \min\left(\frac{1}{4\frac{\max_{i \in [k]} \lambda_i}{\lambda_1} + 2}\lambda_1, \frac{1 - (1+\alpha)/2}{2}\lambda_1\right) \frac{1}{\tau^2 n}, \quad \forall t.$$

Note that $\left|\langle \boldsymbol{v}_1, \tilde{\varepsilon}(\boldsymbol{u}_t)\rangle\right| \leq \frac{1-(1+\alpha)/2}{2\sqrt{2}(1+\alpha)}\lambda_1\frac{1}{\tau^2 n}$ implies the first bound in Eq. (83). In Lemma 4, we assume $\tan\theta\left(\boldsymbol{v}_1, \boldsymbol{u}_0\right) < 1$ and prove that for every $\boldsymbol{u}_t$, $\tan\theta\left(\boldsymbol{v}_1, \boldsymbol{u}_t\right) < 1$, which is equivalent to saying that at every step, $\cos\theta\left(\boldsymbol{v}_1, \boldsymbol{u}_t\right) > \frac{1}{\sqrt{2}}$. By plugging the inequality into the second condition in Lemma 4, we have $\left|\langle\boldsymbol{v}_1, \tilde{\varepsilon}(\boldsymbol{u}_t)\rangle\right| \leq \frac{(\lambda_1 - \lambda_2)}{8}$. The lemma then follows by the fact that $\left\|\left(\boldsymbol{I} - \boldsymbol{u}_T\boldsymbol{u}_T^T\right)\boldsymbol{v}_1\right\| = \sin\theta\left(\boldsymbol{u}_T, \boldsymbol{v}_1\right) \leq \tan\theta\left(\boldsymbol{u}_T, \boldsymbol{v}_1\right) \leq \epsilon$. $\square$

### E.3.2 Deflation

In previous sections we have upper bounded the Euclidean distance between the estimated and the true principal eigenvector of an input tensor $\mathbf{T}$. In this section, we show that error introduced from previous tensor power updates can also be bounded. As a result, we obtain error bounds between the entire set of base vectors $\{\boldsymbol{v}_i\}_{i=1}^{k}$ and their estimation $\{\hat{\boldsymbol{v}}_i\}_{i=1}^{k}$.

**Lemma 6.** *Let $\{\boldsymbol{v}_1, \boldsymbol{v}_2, \cdots, \boldsymbol{v}_k\}$ and $\{\lambda_1, \lambda_2, \cdots, \lambda_k\}$ be orthonormal eigenvectors and eigenvalues of an input tensor $T$. Define $\lambda_{\max} := \max_{i\in[k]}\lambda_i$. Suppose $\{\hat{\boldsymbol{v}}_i\}_{i=1}^{k}$ and $\{\hat{\lambda}_i\}_{i=1}^{k}$ are estimated eigenvector/eigenvalue pairs. Fix $\epsilon \geq 0$ and any $t \in [k]$. If*

$$\left|\hat{\lambda}_i - \lambda_i\right| \leq \lambda_i\epsilon/2, \quad \text{and} \quad \|\hat{\boldsymbol{u}}_i - \boldsymbol{u}_i\| \leq \epsilon \tag{83}$$

*for all $i \in [t]$, then for any unit vector $\boldsymbol{u}$ the following holds:*

$$\left\|\sum_{i=1}^{t}\left[\lambda\boldsymbol{v}_i^{\otimes 3} - \hat{\lambda}_i\hat{\boldsymbol{v}}_i^{\otimes 3}\right](\mathbf{I}, \boldsymbol{u}, \boldsymbol{u})\right\|^2 \leq 4\left(2.5\lambda_{\max} + (\lambda_{\max} + 1.5)\epsilon\right)^2\epsilon^2 + 9(1 + \epsilon/2)^2\lambda_{\max}^2\epsilon^4 \tag{84}$$

$$+ 8(1 + \epsilon/2)^2\lambda_{\max}^2\epsilon^2 \tag{85}$$

$$\leq 50\lambda_{\max}^2\epsilon^2. \tag{86}$$

*Proof.* Following similar approaches in [1], Lemma B.5, we define $\hat{\boldsymbol{v}}^{\perp} = \hat{\boldsymbol{v}}_i - (\boldsymbol{v}_i^{\top}\hat{\boldsymbol{v}}_i)\boldsymbol{v}_i$ and $\mathbf{D}_i = \left[\lambda\boldsymbol{v}_i^{\otimes 3} - \hat{\lambda}_i\hat{\boldsymbol{v}}_i^{\otimes 3}\right]$. $\mathbf{D}_i(\mathbf{I}, \boldsymbol{u}, \boldsymbol{u})$ can then be written as the sum of scaled $\boldsymbol{v}_i$ and $\boldsymbol{v}_i^{\top}$ products as follows:

$$\mathbf{D}_i(\mathbf{I}, \boldsymbol{u}, \boldsymbol{u}) = \lambda_i(\boldsymbol{u}^{\top}\boldsymbol{v}_i)^2\boldsymbol{v}_i - \hat{\lambda}_i(\boldsymbol{u}^{\top}\hat{\boldsymbol{v}}_i)^2\hat{\boldsymbol{v}}_i \tag{87}$$

$$= \lambda_i(\boldsymbol{u}^{\top}\boldsymbol{v}_i)^2\boldsymbol{v}_i - \hat{\lambda}_i(\boldsymbol{u}^{\top}\left(\hat{\boldsymbol{v}}_i^{\perp} + (\boldsymbol{v}_i^{\top}\hat{\boldsymbol{v}}_i)\boldsymbol{v}_i\right))^2\left(\hat{\boldsymbol{v}}^{\perp} + (\boldsymbol{v}_i^{\top}\hat{\boldsymbol{v}}_i)\boldsymbol{v}_i\right) \tag{88}$$

$$= \left(\left(\lambda_i - \hat{\lambda}_i(\boldsymbol{v}_i^{\top}\hat{\boldsymbol{v}}_i)^3\right)(\boldsymbol{u}^{\top}\boldsymbol{v}_i)^2 - 2\hat{\lambda}_i(\boldsymbol{u}^{\top}\hat{\boldsymbol{v}}_i^{\perp})(\boldsymbol{v}_i^{\top}\hat{\boldsymbol{v}}_i)^2(\boldsymbol{u}^{\top}\boldsymbol{v}_i) - \hat{\lambda}_i(\boldsymbol{v}_i^{\top}\hat{\boldsymbol{v}}_i)(\boldsymbol{u}^{\top}\hat{\boldsymbol{v}}^{\perp})\right)\boldsymbol{v}_i$$

$$- \hat{\lambda}_i\left\|\hat{\boldsymbol{v}}_i^{\perp}\right\|\left((\boldsymbol{u}^{\top}\boldsymbol{v}_i)(\boldsymbol{v}_i^{\top}\hat{\boldsymbol{v}}_i) + \boldsymbol{u}^{\top}\hat{\boldsymbol{v}}_i^{\perp}\right)\left(\hat{\boldsymbol{v}}_i^{\perp}/\left\|\hat{\boldsymbol{v}}_i^{\perp}\right\|\right) \tag{89}$$

Suppose $A_i$ and $B_i$ are coefficients of $\boldsymbol{v}_i$ and $\left(\hat{\boldsymbol{v}}_i^{\perp}/\left\|\hat{\boldsymbol{v}}_i^{\perp}\right\|\right)$, respectively. The summation of $\mathbf{D}_i$ can be bounded as

$$\left\|\sum_{i=1}^{t}\mathbf{D}_i(\mathbf{I}, \boldsymbol{u}, \boldsymbol{u})\right\|^2 = \left\|\sum_{i=1}^{t}A_i\boldsymbol{v}_i - \sum_{i=1}^{t}B_i\left(\hat{\boldsymbol{v}}_i^{\perp}/\left\|\hat{\boldsymbol{v}}_i^{\perp}\right\|\right)\right\|_2^2$$

$$\leq 2\left\|\sum_{i=1}^{t}A_i\boldsymbol{v}_i\right\|^2 + 2\left\|\sum_{i=1}^{t}B_i\left(\hat{\boldsymbol{v}}_i^{\perp}/\left\|\hat{\boldsymbol{v}}_i^{\perp}\right\|\right)\right\|^2$$

$$\leq \sum_{i=1}^{t}A_i^2 + 2\left(\sum_{i=1}^{t}|B_i|\right)^2$$

We then try to upper bound $|A_i|$.

$$|A_i| \leq \left|\left(\lambda_i - \hat{\lambda}_i(\boldsymbol{v}_i^{\top}\hat{\boldsymbol{v}}_i)^3\right)(\boldsymbol{u}^{\top}\boldsymbol{v}_i)^2 - 2\hat{\lambda}_i(\boldsymbol{u}^{\top}\hat{\boldsymbol{v}}_i^{\perp})(\boldsymbol{v}_i^{\top}\hat{\boldsymbol{v}}_i)^2(\boldsymbol{u}^{\top}\boldsymbol{v}_i) - \hat{\lambda}_i(\boldsymbol{v}_i^{\top}\hat{\boldsymbol{v}}_i)(\boldsymbol{u}^{\top}\hat{\boldsymbol{v}}^{\perp})\right| \tag{90}$$

$$\leq \left(\lambda_i\left|1 - (\boldsymbol{v}_i^{\top}\hat{\boldsymbol{v}}_i)^3\right| + \left|\lambda_i - \hat{\lambda}_i\right|(\boldsymbol{v}_i^{\top}\hat{\boldsymbol{v}}_i)^3\right)(\boldsymbol{u}^{\top}\boldsymbol{v}_i)^2 + 2\left(\lambda_i + \left|\lambda_i - \hat{\lambda}_i\right|\right)\|\hat{\boldsymbol{v}}_i - \boldsymbol{v}_i\||\boldsymbol{u}^{\top}\boldsymbol{v}_i|$$

$$+ \left(\lambda_i + \left|\lambda_i - \hat{\lambda}_i\right|\right)\|\hat{\boldsymbol{v}}_i - \boldsymbol{v}_i\|^2 \tag{91}$$

$$\leq \left(1.5 \left\|\boldsymbol{v}_i - \hat{\boldsymbol{v}}_i\right\|^2 + \left|\lambda_i - \hat{\lambda}_i\right| + 2\left(\lambda_i + \left|\lambda_i - \hat{\lambda}_i\right|\right)\left\|\boldsymbol{v}_i - \hat{\boldsymbol{v}}_i\right\|\right)\left|\boldsymbol{u}^\top \boldsymbol{v}_i\right|$$
$$+ \left(\lambda_i + \left|\lambda_i - \hat{\lambda}_i\right|\right)\left\|\hat{\boldsymbol{v}}_i - \boldsymbol{v}_i\right\|^2 \tag{92}$$
$$\leq (2.5\lambda_i + (\lambda_i + 1.5)\epsilon)\,\epsilon\left|\boldsymbol{u}^\top \boldsymbol{v}_i\right| + (1 + \epsilon/2)\lambda_i\epsilon^2 \tag{93}$$

Next, we bound $|B_i|$ in a similar manner.

$$|B_i| = \left|\hat{\lambda}_i\right|\left\|\hat{\boldsymbol{v}}_i^\perp\right\|\left|\left((\boldsymbol{u}^\top \boldsymbol{v}_i)(\boldsymbol{v}_i^\top \hat{\boldsymbol{v}}_i) + \boldsymbol{u}^\top \hat{\boldsymbol{v}}_i^\perp\right)\right| \tag{94}$$

$$\leq 2\left(\lambda_i + \left|\lambda_i - \hat{\lambda}_i\right|\right)\left\|\hat{\boldsymbol{v}}_i^\perp\right\|\left((\boldsymbol{u}^\top \boldsymbol{v}_i)^2 + \left\|\hat{\boldsymbol{v}}_i^\perp\right\|^2\right) \tag{95}$$

$$\leq 2(1 + \epsilon/2)\lambda_i\epsilon(\boldsymbol{u}^\top \boldsymbol{v}_i)^2 + 2(1 + \epsilon/2)\lambda_i\epsilon^3 \tag{96}$$

$\square$

Combining everything together we have

$$\left\|\sum_{i=1}^{t}\mathbf{D}_i\left(\mathbf{I}, \boldsymbol{u}, \boldsymbol{u}\right)\right\|^2 \leq 2\sum_{i=1}^{t}A_i^2 + 2\left(\sum_{i=1}^{t}|B_i|\right)^2 \tag{97}$$

$$\leq \sum_{i=1}^{t}4\left(5\lambda_i + (\lambda_i + 1.5)\right)^2\epsilon^2\left|\boldsymbol{u}^\top \boldsymbol{v}_i\right|^2 + 4(1 + \epsilon/2)^2\lambda_i^2\epsilon^4$$

$$+ 2\left(\sum_{i=1}^{t}2(1 + \epsilon/2)\lambda_i\epsilon(\boldsymbol{u}^\top \boldsymbol{v}_i)^2 + 2(1 + \epsilon/2)\lambda_i\epsilon^3\right)^2 \tag{98}$$

$$\leq 4\left(2.5\lambda_{\max} + (\lambda_{\max} + 1.5)\epsilon\right)^2\epsilon^2\sum_{i=1}^{t}\left|\boldsymbol{u}^\top \boldsymbol{v}_i\right|^2 + 4(1 + \epsilon/2)^2\lambda_{\max}^2\epsilon^4$$

$$+ 2\left(2(1 + \epsilon/2)\lambda_{\max}\epsilon\sum_{i=1}^{t}(\boldsymbol{u}^\top \boldsymbol{v}_i)^2 + 2(1 + \epsilon/2)\lambda_{\max}\epsilon^3\right)^2 \tag{99}$$

$$\leq 4\left(2.5\lambda_{\max} + (\lambda_{\max} + 1.5)\epsilon\right)^2\epsilon^2 + 9(1 + \epsilon/2)^2\lambda_{\max}^2\epsilon^4 + 8(1 + \epsilon/2)^2\lambda_{\max}^2\epsilon^2. \tag{100}$$

### E.3.3 Main Theorem

In this section we present and prove the main theorem that bounds the reconstruction error of fast robust tensor power method under appropriate settings of the hash length $b$ and number of independent hashes $B$. The theorem presented below is a more detailed version of Theorem 2 presented in Section 4.2.

**Theorem 3.** *Let* $\bar{\mathbf{T}} = \mathbf{T} + \mathbf{E} \in \mathrm{R}^{n \times n \times n}$, *where* $\mathbf{T} = \sum_{i=1}^{k}\lambda_i\boldsymbol{v}_i^{\otimes 3}$ *and* $\{\boldsymbol{v}_i\}_{i=1}^{k}$ *is an orthonormal basis. Suppose* $(\hat{\boldsymbol{v}}_1, \hat{\lambda}_1), (\hat{\boldsymbol{v}}_1, \hat{\lambda}_1), \cdots (\hat{\boldsymbol{v}}_k, \hat{\lambda}_k)$ *is the sequence of estimated eigenvector/eigenvalue pairs obtained using the fast robust tensor power method. Assume* $\|\mathbf{E}\| = \epsilon$. *There exists constant* $C_1, C_2, C_3, \alpha, \rho, \tau \geq 0$ *such that the following holds: if*

$$\epsilon \leq C_1\frac{1}{n\lambda_{\max}}, \quad and \quad T = C_2\left(\log_{1+\alpha}\left(1 + \rho\right)\tau\sqrt{n} + \frac{\lambda_1}{\lambda_1 - \lambda_2}\log(1/\epsilon)\right), \tag{101}$$

*and*

$$\sqrt{\frac{\ln(L/\log_2(k/\eta))}{\ln(k)}} \cdot \left(1 - \frac{\ln\left(\ln L/\log_2(k/\eta)\right) + C_3}{4\ln\left(L/\log_2(k/\eta)\right)} - \sqrt{\frac{\ln(8)}{\ln(L/\log_2(k/\eta))}}\right) \geq 1.02\left(1 + \sqrt{\frac{\ln(4)}{\ln(k)}}\right). \tag{102}$$

*Suppose the tensor sketch randomness is independent among all tensor product evaluations. If $B = \Omega(\log(n/\tau))$ and the hash length $b$ is set to*

$$b \geq \left\{ \frac{\|\mathbf{T}\|_F^2 \, \tau^4 n^2}{\min\left(\frac{1}{4\max_{i\in[k]}(\lambda_i/\lambda_1)+2}\lambda_1, \frac{1-(1+\alpha)/2}{2\sqrt{2}(1+\alpha)}\lambda_1\right)^2}, \frac{16\epsilon^{-2}\|\mathbf{T}\|_F^2}{\min_{i\in[k]}(\lambda_i-\lambda_{i-1})^2}, \epsilon^{-2}\|\mathbf{T}\|_F^2 \right\} \quad (103)$$

*with probability at least $1 - (\eta + \tau^{-1} + e^{-n})$, there exists a permutation $\pi$ on $k$ such that*

$$\left\|\boldsymbol{v}_{\pi(j)} - \hat{\boldsymbol{v}}_i\right\| \leq \epsilon, \quad \left|\lambda_{\pi(j)} - \hat{\lambda}_j\right| \leq \frac{\lambda_{\pi(j)}\epsilon}{2}, \quad and \quad \left\|\mathbf{T} - \sum_{j=1}^k \hat{\lambda}_j \hat{\boldsymbol{v}}_j^{\otimes 3}\right\| \leq c\epsilon, \quad (104)$$

*for some absolute constant c.*

*Proof.* We prove that at the end of each iteration $i \in [k]$, the following conditions hold

- 1. For all $j \leq i$, $\left|\boldsymbol{v}_{\pi(j)} - \hat{\boldsymbol{v}}_j\right| \leq \epsilon$ and $\left|\lambda_{\pi(j)} - \hat{\lambda}_j\right| \leq \frac{\lambda_i\epsilon}{2}$

- 2. The tensor error satisfies

$$\left\|\left[\left(\tilde{\mathbf{T}} - \sum_{j\leq i}\hat{\lambda}_j\hat{\boldsymbol{v}}_j^{\otimes 3}\right) - \sum_{j\geq i+1}\lambda_{\pi(j)}v_{\pi(j)}^{\otimes 3}\right](\mathbf{I}, \boldsymbol{u}, \boldsymbol{u})\right\| \leq 56\epsilon \quad (105)$$

First, we check the case when $i = 0$. For the tensor error, we have

$$\left\|\left[\tilde{\mathbf{T}} - \sum_{j=1}^K\lambda_{\pi(j)}v_{\pi(j)}^{\otimes 3}\right](\mathbf{I}, \boldsymbol{u}, \boldsymbol{u})\right\| = \|\boldsymbol{\varepsilon}(\boldsymbol{u})\| \leq \|\boldsymbol{\varepsilon}_{2,T}(\boldsymbol{u})\| + \|\mathbf{E}(\mathbf{I}, \boldsymbol{u}, \boldsymbol{u})\| \leq \epsilon + \epsilon = 2\epsilon. \quad (106)$$

The last inequality follows Theorem 1 with the condition for $b$. Next, Using Lemma 5, we have that

$$\left\|\boldsymbol{v}_{\pi(1)} - \hat{\boldsymbol{v}}_1\right\| \leq \epsilon. \quad (107)$$

In addition, conditions for hash length $b$ and Theorem 1 yield

$$\left|\lambda_{\pi(1)} - \hat{\lambda}_1\right| \leq \|\boldsymbol{\varepsilon}_{1,T}(\boldsymbol{v}_1)\| + \|\mathbf{T}(\hat{\boldsymbol{v}}_1 - \boldsymbol{v}_1, \hat{\boldsymbol{v}}_1 - \boldsymbol{u}, \hat{\boldsymbol{v}}_1 - \boldsymbol{v}_1)\| \leq \epsilon\frac{\lambda_i - \lambda_{i-1}}{4} + \epsilon^3\|\mathbf{T}\|_F \leq \frac{\epsilon\lambda_i}{2} \quad (108)$$

Thus, we have proved that for $i = 1$ both conditions hold. Assume the conditions hold up to $i = t-1$ by induction. For the $t$th iteration, the following holds:

$$\left\|\left[\left(\tilde{\mathbf{T}} - \sum_{j\leq t}\hat{\lambda}_j\hat{\boldsymbol{v}}_j^{\otimes 3}\right) - \sum_{j\geq t+1}\lambda_{\pi(j)}v_{\pi(j)}^{\otimes 3}\right](\mathbf{I}, \boldsymbol{u}, \boldsymbol{u})\right\|$$

$$\leq \left\|\left[\tilde{\mathbf{T}} - \sum_{j=1}^K\lambda_{\pi(j)}v_{\pi(j)}^{\otimes 3}\right](\mathbf{I}, \boldsymbol{u}, \boldsymbol{u})\right\| + \left\|\sum_{j=1}^t\hat{\lambda}_j\hat{\boldsymbol{v}}_j^{\otimes 3} - \lambda_{\pi(j)}v_{\pi(j)}^{\otimes 3}\right\| \leq \epsilon + \sqrt{50}\lambda_{\max}\epsilon.$$

For the last inequality we apply Lemma 6. Since the condition is satisfied, Lemma 5 yields

$$\left\|\boldsymbol{v}_{\pi(t+1)} - \hat{\boldsymbol{v}}_{t+1}\right\| \leq \epsilon. \quad (109)$$

Finally, conditions for hash length $b$ and Theorem 1 yield

$$\left|\lambda_{\pi(t+1)} - \hat{\lambda}_{t+1}\right| \leq \|\boldsymbol{\varepsilon}_{1,T}(\boldsymbol{v}_1)\| + \|\mathbf{T}(\hat{\boldsymbol{v}}_t - \boldsymbol{v}_1, \hat{\boldsymbol{v}}_1 - \boldsymbol{u}, \hat{\boldsymbol{v}}_1 - \boldsymbol{v}_1)\|$$

$$\leq \epsilon\frac{\lambda_i - \lambda_{i-1}}{4} + \epsilon^3\|\mathbf{T}\|_F \leq \frac{\epsilon\lambda_i}{2} \quad (110)$$

$\square$

## Appendix F  Summary of notations for matrix/vector products

We assume vectors $\boldsymbol{a}, \boldsymbol{b} \in \mathbb{C}^n$ are indexed starting from 0; that is, $\boldsymbol{a} = (a_0, a_1, \cdots, a_{n-1})$ and $\boldsymbol{b} = (b_0, b_1, \cdots, b_{n-1})$. Matrices $\mathbf{A}, \mathbf{B}$ and tensors $\mathbf{T}$ are still indexed starting from 1.

**Element-wise product**    For $\boldsymbol{a}, \boldsymbol{b} \in \mathbb{C}^n$, the element-wise product (Hadamard product) $\boldsymbol{a} \circ \boldsymbol{b} \in \mathbb{R}^n$ is defined as

$$\boldsymbol{a} \circ \boldsymbol{b} = (a_0 b_0, a_1 b_1, \cdots, a_{n-1} b_{n-1}). \tag{111}$$

**Convolution**    For $\boldsymbol{a}, \boldsymbol{b} \in \mathbb{C}^n$, their convolution $\boldsymbol{a} * \boldsymbol{b} \in \mathbb{C}^n$ is defined as

$$\boldsymbol{a} * \boldsymbol{b} = \left( \sum_{(i+j) \mod n = 0} a_i b_j, \sum_{(i+j) \mod n = 1} a_i b_j, \cdots, \sum_{(i+j) \mod n = n-1} a_i b_j \right). \tag{112}$$

**Inner product**    For $\boldsymbol{a}, \boldsymbol{b} \in \mathbb{C}^n$, their inner product is defined as

$$\langle \boldsymbol{a}, \boldsymbol{b} \rangle = \sum_{i=1}^{n} a_i \overline{b_i}, \tag{113}$$

where $\overline{b_i}$ denotes the complex conjugate of $b_i$. For tensors $\mathbf{A}, \mathbf{B} \in \mathbb{C}^{n \times n \times n}$, their inner product is defined similarly as

$$\langle \mathbf{A}, \mathbf{B} \rangle = \sum_{i,j,k=1}^{n} \mathbf{A}_{i,j,k} \overline{\mathbf{B}}_{i,j,k}. \tag{114}$$

**Tensor product**    For $\boldsymbol{a}, \boldsymbol{b} \in \mathbb{C}^n$, the tensor product $\boldsymbol{a} \otimes \boldsymbol{b}$ can be either an $n \times n$ matrix or a vector of length $n^2$. For the former case, we have

$$\boldsymbol{a} \otimes \boldsymbol{b} = \begin{bmatrix} a_0 b_0 & a_0 b_1 & \cdots & a_0 b_{n-1} \\ a_1 b_0 & a_1 b_1 & \cdots & a_1 b_{n-1} \\ \vdots & \vdots & \ddots & \vdots \\ a_{n-1} b_0 & a_{n-1} b_1 & \cdots & a_{n-1} b_{n-1} \end{bmatrix}. \tag{115}$$

If $\boldsymbol{a} \otimes \boldsymbol{b}$ is a vector, it is defined as the expansion of the output matrix. That is,

$$\boldsymbol{a} \otimes \boldsymbol{b} = (a_0 b_0, a_0 b_1, \cdots, a_0 b_{n-1}, a_1 b_0, a_1 b_1, \cdots, a_{n-1} b_{n-1}). \tag{116}$$

Suppose $\mathbf{T}$ is an $n \times n \times n$ tensor and matrices $\mathbf{A} \in \mathbb{R}^{n \times m_1}$, $\mathbf{B} \in \mathbb{R}^{n \times m_2}$ and $\mathbf{C} \in \mathbb{R}^{n \times m_3}$. The tensor product $\mathbf{T}(\mathbf{A}, \mathbf{B}, \mathbf{C})$ is an $m_1 \times m_2 \times m_3$ tensor defined by

$$[\mathbf{T}(\mathbf{A}, \mathbf{B}, \mathbf{C})]_{i,j,k} = \sum_{i',j',k'=1}^{n} \mathbf{T}_{i',j',k'} \mathbf{A}_{i',i} \mathbf{B}_{j',j} \mathbf{C}_{k',k}. \tag{117}$$

**Khatri-Rao product**    For $\mathbf{A}, \mathbf{B} \in \mathbb{C}^{n \times m}$, their Khatri-Rao product $\mathbf{A} \odot \mathbf{B} \in \mathbb{C}^{n^2 \times m}$ is defined as

$$\mathbf{A} \odot \mathbf{B} = (\mathbf{A}_{(1)} \otimes \mathbf{B}_{(1)}, \mathbf{A}_{(2)} \otimes \mathbf{B}_{(2)}, \cdots, \mathbf{A}_{(m)} \otimes \mathbf{B}_{(m)}), \tag{118}$$

where $\mathbf{A}_{(i)}$ and $\mathbf{B}_{(i)}$ denote the $i$th rows of $\mathbf{A}$ and $\mathbf{B}$.

**Mode expansion**    For a tensor $\mathbf{T}$ of dimension $n \times n \times n$, its first mode expansion $\mathbf{T}_{(1)} \in \mathbb{R}^{n \times n}$ is defined as

$$\mathbf{T}_{(1)} = \begin{bmatrix} \mathbf{T}_{1,1,1} & \mathbf{T}_{1,1,2} & \cdots & \mathbf{T}_{1,1,n} & \mathbf{T}_{1,2,1} & \cdots & \mathbf{T}_{1,n,n} \\ \mathbf{T}_{2,1,1} & \mathbf{T}_{2,1,2} & \cdots & \mathbf{T}_{2,1,n} & \mathbf{T}_{2,2,1} & \cdots & \mathbf{T}_{2,n,n} \\ \vdots & \vdots & \vdots & \vdots & \vdots & \vdots & \vdots \\ \mathbf{T}_{n,1,1} & \mathbf{T}_{n,1,2} & \cdots & \mathbf{T}_{n,1,n} & \mathbf{T}_{n,2,1} & \cdots & \mathbf{T}_{n,n,n} \end{bmatrix}. \tag{119}$$

The mode expansions $\mathbf{T}_{(2)}$ and $\mathbf{T}_{(3)}$ can be similarly defined.

## Footnotes

[5]As long as $\mathbf{A}$ is symmetric, we have $\langle \mathbf{A}, \mathbf{Y}_i \rangle = \langle \mathbf{A}, \mathbf{Z}_i \rangle / 3$.

[6] For a tensor $\mathbf{T} \in \mathbb{R}^{V \times V \times V}$ and a matrix $\mathbf{W} \in \mathbb{R}^{V \times k}$, the product $\mathbf{Q} = \mathbf{T}(\mathbf{W}, \mathbf{W}, \mathbf{W}) \in \mathbb{R}^{k \times k \times k}$ is defined as $\mathbf{Q}_{i_1,i_2,i_3} = \sum_{j_1,j_2,j_3=1}^V \mathbf{T}_{j_1,j_2,j_3}\mathbf{W}_{j_1,i_1}\mathbf{W}_{j_2,i_2}\mathbf{W}_{j_3,i_3}$.

[7] and also $\hat{\mathbb{E}}[\boldsymbol{x}_1 \otimes \widehat{\mathbf{M}}_1 \otimes \boldsymbol{x}_2](\mathbf{W}, \mathbf{W}, \mathbf{W})$, $\hat{\mathbb{E}}[\widehat{\mathbf{M}}_1 \otimes \boldsymbol{x}_1 \otimes \boldsymbol{x}_2](\mathbf{W}, \mathbf{W}, \mathbf{W})$ by symmetry.

[8] Note that $\text{sum}(\mathcal{M}(\boldsymbol{l})) = \text{sum}(\mathcal{M}(\boldsymbol{l}'))$ and hence $\|\mathcal{M}(\boldsymbol{l}) - \mathcal{M}(\boldsymbol{l}')\|_1$ must be even. Furthermore, the sum of positive entries in $(\mathcal{M}(\boldsymbol{l}) - \mathcal{M}(\boldsymbol{l}'))$ equals the sum of negative entries.