[Reviews · NeurIPS 2015]

Submitted by Assigned_Reviewer_1

# Summary The paper applies existing tensor sketching ideas to orthogonal tensor decomposition, leading to an algorithm with running time O(n^2 log(n)) (vs. O(n^3)), when the eigenvalue spectrum decays superlinearly. The authors propose a new hashing method for symmetric tensors, and validate their computational speedups on topic models.

# Comments * The idea of storing a low-dimensional sketched representation is exciting, however the results presented in Theorem 2 only hold when the spectrum of the tensor (the eigenvalues) decay super-linearly, as in the synthetic experiments (e.g. $\lambda_i = 1/i$). In the case where the spectrum is flat, i.e. $\lambda_i$ are approximately the same, the size of the sketch required is $O(n^3)$, according to Theorem 2. Is this an artifact of the analysis or a limitation of the approach? * In the experiments, and in table 3, the sketch building time is ignored -- what exactly comprises of sketch building time? It seems like this is a crucial component to the method. * How does the method scale with the magnitude of error ($\sigma$)? * Is the sketching of a rank-1 tensor (3.2.1) a contribution of the current paper? It appears to be presented in [22; Algorithm 1]. * How does the symmetric tensor sketch proposed in 3.4 differ from the idea of using the complex ring to count higher moments, e.g. Ganguly, Sumit. "Estimating frequency moments of data streams using random linear combinations." Approximation, Randomization, and Combinatorial Optimization. Algorithms and Techniques. Springer Berlin Heidelberg, 2004. 369-380.

# Minor commments * The notation in the paper is hard to follow and very densely packed. In some places, the notation is not clearly defined, e.g. 3.1., [n] was not defined, and $p$ is not described. * In Algorithm 2, the footnote 4 is easily confused with an exponent. Also, it is not clear whether line 7 has a typo: "$[v^{(m)}]_i \gets \ksi_1(i) ..$" <- is it always $\ksi_1$? * What is meant by time constraints when describing the synthetic experiments (margin line #392).
Summary: The paper presents a useful contribution to the tensor decomposition literature. It would merit from a clearer exposition and better notation. Some details of the experimental setup should be clarified.

Submitted by Assigned_Reviewer_2

The authors observe that a rank-1 tensor sketch can be expressed as a simple element-wise multiplication of vectors under an FFT transform (3). By exploiting the linearity of the sketching operator, they present a fast sketching method for rank-N tensors that works directly on factored moments (Algorithm 1). They use this sketching scheme (plus a trick to avoid repeated computations) to attack the computational bottleneck in the robust tensor power method of Anandkumar et al. - computing T(u,u,u) and T(I,u,u) (Algorithm 2). Furthermore, they design a "colliding" hash tailored for symmetric tensors, which requires an extension of the original Rademacher random variables to the complex domain. Finally, they provide concentration bounds on the construction and decomposition of tensors via the proposed sketching method (Theorem 1 and 2).

As can be seen in the summary, there's a lot going on in this paper. The novel sketching method is well posed both theoretically and empirically. I haven't read the supplementary materials but the authors also apply their sketching scheme to other CP decomposition methods such as ALS, which makes the paper even more impressive. I think the paper is a clear accept.

It might be worth looking into the tensor decomposition method of Kuleshov et al. to see if the sketching method is useful there as well. In that method, you perform a series of random contractions from tensors to matrices and do simultaneous diagonalization. At least the contraction part may be able to benefit from sketching.

Summary: The paper proposes a novel sketching method for tensors based on an FFT transform, which can be naturally incorporated in important tasks such as tensor decomposition. The paper makes several concrete contributions along the way which shed light on the problem.

Submitted by Assigned_Reviewer_3

Summary: In this paper authors propose a new sketching algorithm to compute tensor factorization. The algorithm is based on tensor sketch of Pham and Pagh.

For the case of symmetric factor tensor authors propose new ways to compute sketch fast.

Given a tensor (either entries or as factors), algorithm computes the sketch and uses the sketch to compute eigen vectors of the tensor using power method or ALS appropriately modified to use the sketches. Authors provide guarantees for the case when the input is orthogonal tensor, on the sketch sizes and the corresponding error rate. Finally they show performance of this algorithm through experiments on both real and synthetic data.

Comments:

1. Authors claim in related work that whitening based dimensionality reduction is slow in practice even though it has much lower per iteration complexity compared to the proposed sketching method. Authors should substantiate this claim either through experiments or by citing appropriate references.

2. For the case when input is factor vectors and not the entries of the higher order tensor, why not just use simple tensor power method on the factors directly? This does not need to form the higher order tensor explicitly. For N factors in n dimensions, each iteration of tensor power method takes only O(n *N) time and memory. The proposed method has a complexity of O(N *(n + blog(b))) +O((n + blog(b))) for subsequent iterations.

Since b is Omega(n^2) from Theorem 4.2 the sketching technique for this case seems to be of higher complexity for N ~O(n) which is common in practice. Please add a clear discussion for the factor case.

2. b is atleast O(n^2) according to Thm 4.2, assuming \delta and r(\lambda) are constant. This seems to be weaker than what is needed in simulations. Please comment on this.

3. \lambda computation is missing in algorithm 2.
Summary: In this paper authors propose a new one pass sketching technique to speed up tensor power method from O(n^3) to O(n + blog(b)) ~ O(n^2) per iteration time. The method has good practical performance and error guarantees.

However it is not compared well with the existing sketching techniques and the application to the factored case is not convincing.

Author Feedback
Author rebuttal: We thank the reviewers for their helpful comments.

To R#1:
- Our analysis does require the eigenvalues to have at least a moderate decay. We believe it is an artifact of our analysis since we used a naive step-by-step deflation analysis. A more refined blockwise type analysis similar to http://arxiv.org/abs/1311.2495 could potentially circumvent this assumption. In addition, eigenvalues of real-world data tensors usually follow a power-law distribution.

- The sketching build time is O(nnz(T)) for general tensors and O(n+blog b) for low-rank tensors. For latent variable models sketches can be built efficiently. Also, we included sketch building time in all real-world experiments (Fig 1, 2, Tab 3, etc.)

- The noise level sigma does not affect the running time of our proposed method at all. It affects the accuracy of recovered eigenvectors.

- The procedure in Sec 3.2.1 is from [22]. We will make this clear in the final version of the paper. The empirical moment extension (Sec 3.2.2) and colliding hash design (Sec 3.4) are novel.

- Comparison to the idea of using complex rings for moment estimation: we thank the reviewer for pointing us to this paper. The paper deals with estimation of frequency moments, which are aggregated over the different items, which is a different problem. Our problem deals with sketching general tensors.

To R#2:
We thank the reviewer for pointing us to Kuleshov et al.'s paper. We agree that the tensor contraction part in Kuleshov et al.'s method may also be accelerated by sketching and we will look into that in future work.

To R#3:
- We agree that per-iteration time for online tensor decomposition methods might be shorter and could run faster when N is small. However, tensor decomposition usually makes many iterations, which renders the total running time of online tensor decomposition slow. In addition, in typical applications N is overwhelmingly larger than n. For example, in topic modeling the number of topics (n) is usually from 100 to 1000 but the number of documents (N) could easily be a million or even a billion. Last but not the least, the online tensor decomposition method requires going through training data many times, which may cause difficult engineering and data storage problems. We will add these comments in the final version of the paper.

- We do think the error bounds in Theorem 2 are loose and experimental results show strong performance of our proposed algorithm. We are working to come up with a more refined theoretical analysis.

- The procedure of estimating lambda is to perform another approximate T(u, u, u) computation. Thanks for pointing out this typo.

To R#4, R#5 and R#6:
Thanks for reading and reviewing our paper. We will continue working on improving the paper.